# CoPRS: Learning Positional Prior from Chain-of-Thought for Reasoning Segmentation

**Zhenyu Lu**[1,2,5]**, Liupeng Li**[3,2]**, Jinpeng Wang**[3,*]**, Yan Feng**[4]**, Bin Chen**[3]**, Ke Chen**[2]**, Yaowei Wang**[3,2,*]

[1]Shenzhen Institutes of Advanced Technology, Chinese Academy of Sciences
[2]Peng Cheng Laboratory
[3]Harbin Institute of Technology, Shenzhen
[4]Meituan, Beijing
[5]University of Chinese Academy of Sciences
`zhenyulu22@m.fudan.edu.cn; wangjp26@gmail.com; wangyaowei@hit.edu.cn`

## Abstract

Existing works on reasoning segmentation either connect hidden features from a language model directly to a mask decoder or represent positions in text, which limits interpretability and semantic detail. To solve this, we present CoPRS, a Multi-modal Chain-of-Thought (MCoT)-based positional perception model that bridges language reasoning to segmentation through a differentiable and interpretable positional prior instantiated as a heatmap. By making the reasoning process clear via MCoT and expressing it as a dense, differentiable heatmap, this interface enhances interpretability and diagnostic analysis and yields more concentrated evidence on the target. A learnable concentration token aggregates features of the image and reasoning text to generate this positional prior, which is decoded to precise masks through a lightweight decoder, providing a direct connection between reasoning and segmentation. Across the RefCOCO series and ReasonSeg, CoPRS matches or surpasses the best reported metrics on each standard split under comparable protocols, with performance at or above the prior state of the art across both validation and test partitions. Extensive experiments demonstrate a strong positive correlation among the CoT trajectory, the generated heatmap, and the decoded mask, supporting an interpretable alignment between the reasoning output and downstream mask generation. Collectively, these findings support the utility of this paradigm in bridging reasoning and segmentation and show advantages in concentration driven by reasoning and in more precise mask prediction. Code has been released at `https://github.com/ZhenyuLU-Heliodore/CoPRS`.

## 1 Introduction

Visual perception is increasingly expected to not only assign labels to pixels but also follow natural-language instructions with compositional constraints, such as "Segment the UAV that is trailing the quadcopter and partially occluded by trees." This demand advances the long arc of visual understanding, starting from semantic segmentation (category labels) (Guo et al., 2018), to instance segmentation (object masks) (Hafiz & Bhat, 2020), and further to open-vocabulary segmentation (open-set text categories) (Ren et al., 2024a), and most recently, toward ***reasoning segmentation*** (free-form instructions) Lai et al. (2024). Meeting this goal requires coupling language reasoning with spatial grounding by converting textual instructions into perceptual decisions.

Existing attempts to bridge language reasoning with segmentation fall into two distinct camps. ***Latent reasoning*** methods (Pi et al., 2024; Lai et al., 2024) predict the masks by directly decoding hidden features from the language models, which keep intermediate decisions non-transparent and uncontrollable. ***Text-based reasoning*** methods (Lan et al., 2025; Liu et al., 2025), on the other hand, readout positions in text and generate discrete coordinates. While explicit, such an interface is inflexible to capture and reflect fine-grained visual semantics, and also fragile to practical issues like

---

*Corresponding authors.

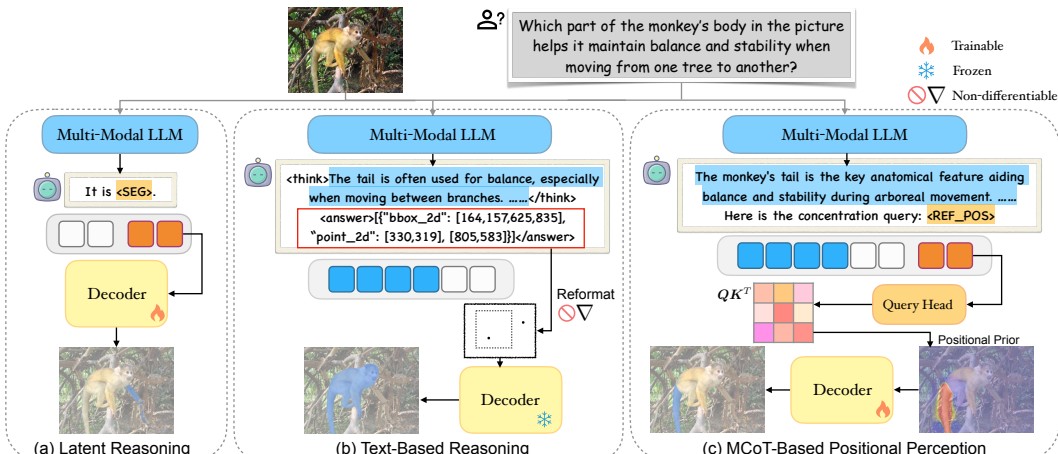

Figure 1: **Illustration of paradigms for reasoning segmentation.** (a) is exemplified by LISA (Lai et al., 2024), and (b) by Seg-Zero (Liu et al., 2025). Our CoPRS (c) bridges MCoT reasoning to segmentation through a differentiable and interpretable positional prior.

formatting errors or out-of-image coordinates. In essence, limitations in the two polarized paradigms highlight the need for a better trade-off between interpretability and representational fidelity.

To close this gap, we introduce **CoPRS**, a **Co**T-based **P**ositional perception model for **R**easoning **S**egmentation. CoPRS is one-stage and end-to-end: given an image–instruction input, it first reasons before producing a perception heatmap concentrating the target region, which provides a *positional prior* to enhance the segmentation mask decoding. As compared in Figure 1, the positional prior serves as a differentiable and interpretable connection between MCoT (Wang et al., 2025b) and segmentation, which is direct and effective to enhance visual perception of a Multi-modal Large Language Model (MLLM) and align instruction semantics with mask decoding.

Specifically, we first introduce a learnable concentration token to aggregate image–instruction context and generate a concentration query. Next, we convert this query the positional prior to concentrate the target for mask prediction. This dense, differentiable heatmap is more interpretable than purely hidden features, and provides finer detail than discrete textual coordinates. Concurrently, we establish a unified training framework by adopting the Group Relative Policy Optimization (GRPO) (Shao et al., 2024) jointly with segmentation supervision. This framework enhances reasoning capability through GRPO, jointly supervising the MLLM and segmentation model via a differentiable positional prior and offering an effective solution to the limitations of prior paradigms.

CoPRS matches or exceeds the best reported cIoU/gIoU on each split under comparable protocols across RefCOCO, RefCOCO+ (Kazemzadeh et al., 2014), RefCOCOg (Mao et al., 2016), and ReasonSeg (Lai et al., 2024). We further find a strong positive correlation among the quality of the CoT trajectory, the generated heatmap, and the decoded mask, indicating strong concentration driven by reasoning and precise mask generation. Beyond reasoning segmentation, the unified framework and its positional prior naturally extend to region concentration tasks such as trajectory prediction.

To summarize, we make the following contributions in this paper.

- **CoPRS Formulation.** We present an end-to-end MCoT-driven positional perception model for reasoning segmentation, where a language-conditioned positional prior serves as an interpretable intermediate aligning instruction understanding with mask prediction.

- **Unified Framework.** We establish a unified training framework by combining a GRPO strategy with a supervised objective, enhancing reasoning and segmentation in a single loop and overcoming the limitations of prior paradigms.

- **Positional Prior Interface.** A learnable concentration query produces a heatmap as a dense positional prior, and a lightweight decoder refines it into a precise mask. Our design provides both interpretable concentration and strong boundary quality.

- **Strong Results.** CoPRS performs strongly on each split across the RefCOCO series and ReasonSeg, and further analysis clarifies how reasoning output aligns with segmentation performance.

## 2 RELATED WORK

**Referring and Reasoning Segmentation.** Referring segmentation requires a model to produce a mask for the entity described in a short instruction. Prior methods such as VLT (Ding et al., 2021), CRIS (Wang et al., 2022), LAVT (Yang et al., 2022), ReLA (Liu et al., 2023a), X-Decoder (Zou et al., 2023a), SEEM (Zou et al., 2023b), CD-ViTO (Fu et al., 2024), Grounded-SAM (Ren et al., 2024a), typically rely on specific text encoders rather than large language models (LLMs) to parse the text and predict the mask. Reasoning segmentation extends this setting to longer, compositional instructions with stricter grounding requirements, motivating the two method families outlined next.

**Latent Reasoning Methods.** Advances in multimodal large language models (MLLMs) (Liu et al., 2023b; Bai et al., 2023) have substantially improved the reasoning capability of vision–language perception. LISA (Lai et al., 2024) bridges the gap between MLLMs and reasoning segmentation by introducing a special token. Subsequent works, including PerceptionGPT (Pi et al., 2024), PixelLM (Ren et al., 2024b), SegLLM (Ren et al., 2024b), LaSagnA (Wei et al., 2024), OMG-LLaVA (Zhang et al., 2024a), GroundHog (Zhang et al., 2024b), ObjectRelator (Fu et al., 2025), RAS (Cao et al., 2025), leverage LLM latent features and decode them into segmentation masks. However, they neither reveal intermediate reasoning before the final prediction nor expose it through a transparent interface. In contrast, our approach makes the reasoning process clear via MCoT and visualizes the intermediate as a heatmap, improving interpretability and diagnostic analysis.

**Text-based Reasoning Methods.** Since SAM (Kirillov et al., 2023) achieves strong segmentation quality when prompted with boxes or points, it is feasible to prompt SAM using textual coordinates after a simple format conversion. Recent works, such as SAM4MLLM (Chen et al., 2024), Seg-Zero (Liu et al., 2025) and Seg-R1 (You & Wu, 2025), use MLLMs to generate textual coordinates of boxes and points via chain-of-thought, and then feed them to SAM for mask prediction. In a similar vein, Text4Seg (Lan et al., 2025) generates textual patch indices and applies CRF (Krähenbühl & Koltun, 2011) or SAM for mask refinement. Such sparse, discrete outputs provide limited semantic detail and are sensitive to formatting errors and out-of-image coordinates. To address these issues, our model introduces a dense, differentiable positional prior that captures richer semantic detail.

## 3 METHOD

We first present the model design and data flow in Section 3.1. We then formalize the learning objectives, unifying policy optimization via GRPO on the language path with segmentation supervision on the vision path in Section 3.2. Finally, we detail the training and inference procedures in Section 3.3, including data preparation, tokenization, group rollouts, and deterministic inference.

### 3.1 MODEL ARCHITECTURE

**Overall Architecture.** As shown in Figure 2, CoPRS is built upon a multimodal LLM (MLLM), a vision backbone, a query head and a mask decoder. Given image and text inputs $(\boldsymbol{x}_{\text{img}}, \boldsymbol{x}_{\text{txt}})$, a policy model $\pi_{\boldsymbol{\theta}}(\cdot)$ generates a token sequence that includes the chain-of-thought (CoT) and a concentration token, and we read the MLLM's hidden states to obtain the concentration token embedding. Then the query head $\mathcal{F}_{\text{head}}(\cdot)$ maps this embedding to a concentration query. The vision encoder $\mathcal{F}_{\text{enc}}(\cdot)$ extracts image features as image keys. Subsequently, the query attends to the image keys with multi-head attention, yielding a heatmap that serves as a positional prior. Finally, the mask decoder $\mathcal{F}_{\text{dec}}(\cdot)$ decodes this prior to the predicted mask $\hat{\boldsymbol{M}}$.

**MLLM Backbone.** We use Qwen2.5-VL (Bai et al., 2025) as our MLLM backbone. Following DeepSeek-R1 (Guo et al., 2025), we adopt multimodal chain-of-thought (MCoT) to leverage the reasoning capabilities of MLLM on compositional instructions. Specifically, we use an instruction prompt to elicit both the CoT and a concentration token: given $(\boldsymbol{x}_{\text{img}}, \boldsymbol{x}_{\text{txt}})$, the model is asked to (i) reason in a `<think>...</think>` block and then (ii) output the concentration token `<REF_POS>`. We obtain the concentration token's embedding $\boldsymbol{e}_{\text{conc}}$ via $\mathcal{F}_{\text{conc}}$ which finds its occurrence and reads the hidden states of LLM. Under this setup, the policy $\pi_{\boldsymbol{\theta}}$ generates the token sequence $\boldsymbol{y}_{1:T}$ via next token prediction. Formally, the process is given in

$$
\begin{aligned}
y_t &\sim \pi_{\boldsymbol{\theta}}(\cdot \mid \boldsymbol{y}_{0:t-1}, \boldsymbol{x}_{\text{img}}, \boldsymbol{x}_{\text{txt}}), \; t = 1, \ldots, T, \\
\boldsymbol{e}_{\text{conc}} &= \mathcal{F}_{\text{conc}}(\boldsymbol{y}_{1:T}),
\end{aligned}
\tag{1}
$$

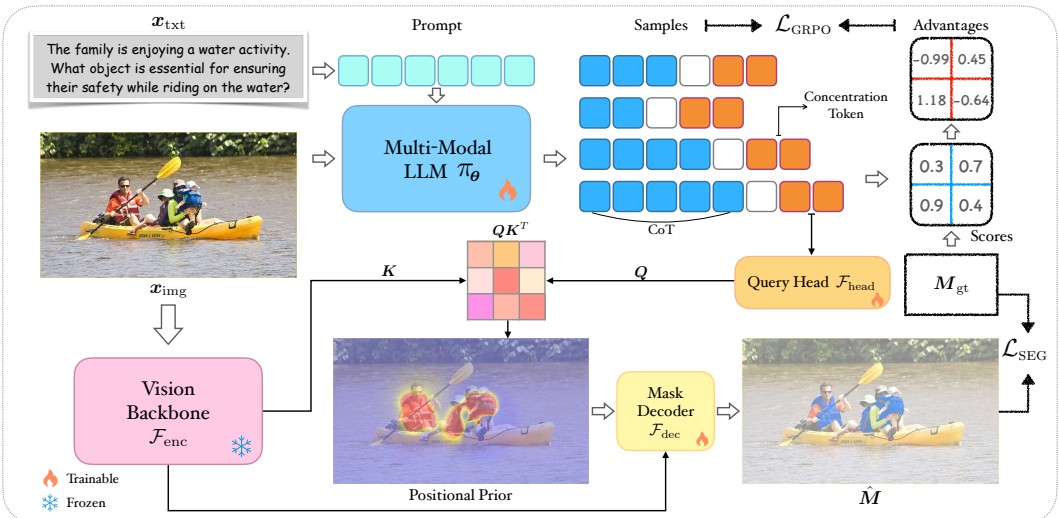

Figure 2: **Overall architecture.** Given image and text inputs, the policy generates CoT and concentration tokens, which query image keys to generate a positional prior, that is then decoded to masks. The policy and segmentation modules are jointly optimized.

where $\boldsymbol{y}_{1:T}$ includes both the CoT and the concentration token.

**From Keys and a Query to Positional Prior.** The vision backbone encodes $\boldsymbol{x}_{\text{img}}$ into image features, which we map to vision keys $\boldsymbol{K}$ via a multilayer perceptron (MLP) applied to the backbone output. In practice, we choose ViT-H — an image encoder from SAM (Kirillov et al., 2023) as the vision backbone and an MLP query head projects $\boldsymbol{e}_{\text{conc}}$ into the concentration query $\boldsymbol{Q}$. Subsequently, we compute scaled dot product multi-head attention scores (Vaswani et al., 2017) between $\boldsymbol{Q}$ and $\boldsymbol{K}$, and we use two stacked 2D convolutional layers denoted $\mathcal{F}_{\text{fuse}}(\cdot)$ to aggregate features across heads. Formally, the computation is defined in the following equations.

$$\boldsymbol{K} = \mathcal{F}_{\text{enc}}(\boldsymbol{x}_{\text{img}}), \quad \boldsymbol{Q} = \mathcal{F}_{\text{head}}(\boldsymbol{e}_{\text{conc}}),$$
$$\boldsymbol{H}_{\text{prior}} = \mathcal{F}_{\text{fuse}}\Big( \big[(\boldsymbol{Q}\boldsymbol{W}_i^Q)(\boldsymbol{K}\boldsymbol{W}_i^K)^\top / \sqrt{d_c}\big]_{i=1}^{n_{\text{head}}} \Big), \tag{2}$$

where $\boldsymbol{Q} \in \mathbb{R}^{d_q}$, $\boldsymbol{K} \in \mathbb{R}^{H \times W \times d_k}$, $\boldsymbol{W}_i^Q \in \mathbb{R}^{d_q \times d_h}$, $\boldsymbol{W}_i^K \in \mathbb{R}^{d_k \times d_h}$, $d_h$ is the head dimension, $n_{\text{head}}$ is the number of heads, and $\mathcal{F}_{\text{fuse}} : \mathbb{R}^{n_{\text{head}} \times H \times W} \to \mathbb{R}^{H \times W}$. Details are provided in Algorithm 1.

**Lightweight Decoder.** Our mask decoder comprises two submodules. First, three stacked 2D convolutional blocks resample the fused positional prior, producing a feature map at the decoder resolution. Second, we choose a Two-Way Transformer following the SAM decoder design (Kirillov et al., 2023), which performs bidirectional cross attention between the image features and the positional prior. This lightweight design has 4.7M parameters and enables the prior to guide dense segmentation. Formally, we formulate the process as

$$\hat{\boldsymbol{M}} = \mathcal{F}_{\text{dec}}(\boldsymbol{K}, \boldsymbol{H}_{\text{prior}}). \tag{3}$$

### 3.2 LEARNING OBJECTIVES

**Unified Objective.** We train the whole system end-to-end with a single objective that couples reinforcement learning on the language path with segmentation supervision on the vision path. For each $(\boldsymbol{x}_{\text{img}}, \boldsymbol{x}_{\text{txt}})$, the policy $\pi_{\boldsymbol{\theta}}$ rolls out a group of responses $\big\{\boldsymbol{y}_{1:T_i}^{(i)}\big\}_{i=1}^G$ with the group size $G$, and we compute a GRPO loss $\mathcal{L}_{\text{GRPO}}$ from the advantages. In parallel, the positional prior $\boldsymbol{H}_{\text{prior}}$ and the predicted mask $\hat{\boldsymbol{M}}$ are supervised against the ground truth mask $\boldsymbol{M}_{\text{gt}}$ to yield the segmentation loss $\mathcal{L}_{\text{SEG}}$. The overall objective is

$$\mathcal{L} = \mathcal{L}_{\text{GRPO}}\Big(\big\{\boldsymbol{y}_{1:T_i}^{(i)}\big\}_{i=1}^G\Big) + \lambda_{\text{SEG}} \mathcal{L}_{\text{SEG}}\Big(\boldsymbol{H}_{\text{prior}}, \hat{\boldsymbol{M}}, \boldsymbol{M}_{\text{gt}}\Big). \tag{4}$$

We compute both terms for each batch and take a single backward pass through all trainable modules.

**GRPO Objective.** Following Shao et al. (2024), we optimize $\pi_{\boldsymbol{\theta}}$ with the GRPO objective. The update ratio $r_{i,t}$ is the likelihood ratio between the current policy $\pi_{\boldsymbol{\theta}}$ and the old policy $\pi_{\boldsymbol{\theta}_{\mathrm{old}}}$ at token $o_{i,t}$, which is clipped with $\varepsilon$ introduced in PPO (Schulman et al., 2017) for stability. The advantage $\widehat{A}_{i,t}$ is computed relative rewards within each group only; details are given in Section A.1. Formally, the policy loss is

$$\mathcal{L}_{\pi} = \mathbb{E}_{i,t}\Big[\min\Big(r_{i,t}\,\widehat{A}_{i,t},\, \mathrm{clip}\big(r_{i,t},\, 1-\varepsilon,\, 1+\varepsilon\big)\,\widehat{A}_{i,t}\Big)\Big], \quad t = 1 : T_i,\; i = 1 : G, \qquad (5)$$

where the update ratio

$$r_{i,t} = \frac{\pi_{\boldsymbol{\theta}}\big(o_{i,t}\,\big|\,\boldsymbol{o}_{i,1:t-1},\,\boldsymbol{x}_{\mathrm{img}},\,\boldsymbol{x}_{\mathrm{txt}}\big)}{\pi_{\boldsymbol{\theta}_{\mathrm{old}}}\big(o_{i,t}\,\big|\,\boldsymbol{o}_{i,1:t-1},\,\boldsymbol{x}_{\mathrm{img}},\,\boldsymbol{x}_{\mathrm{txt}}\big)}, \qquad (6)$$

and the token $o_{i,t} = y_t^{(i)}$. GRPO further regularizes with a KL divergence term between the trained policy and the reference policy:

$$\mathcal{L}_{\mathrm{GRPO}} = \mathcal{L}_{\pi} - \beta\,\mathbb{D}_{\mathrm{KL}}\big[\,\pi_{\boldsymbol{\theta}}\,\|\,\pi_{\mathrm{ref}}\,\big], \qquad (7)$$

where $\beta$ is the coefficient of the KL penalty (See Section A.1).

For each sampled response in the group, we design a reward function that combines mask quality and CoT format compliance. Specifically, the mask reward score aggregates soft IoU, soft dice, and hard IoU, while the CoT format reward score is computed via multiple regular expressions for the string matching. We then normalize both rewards to the range $[0, 1]$ using fixed coefficients. Further implementation details are provided in Section 4.1.

**Supervised Segmentation Objective.** The segmentation loss comprises three complementary terms. (i) A binary cross-entropy (BCE) loss applied to $\boldsymbol{H}_{\mathrm{prior}}$ encourages positional evidence and accurate concentration. (ii) A dice loss (Milletari et al., 2016) on the predicted mask $\hat{\boldsymbol{M}}$ directly supervises mask quality. (iii) A focal loss (Lin et al., 2017) on the mask logits emphasizes hard pixels and fine-grained structures. All losses are computed only over the original image region and averaged per image over the batch, with the dice loss coefficient $\lambda_d$ and focal loss coefficient $\lambda_f$ being reported in Section 4.1. Formally, the segmentation loss is

$$\mathcal{L}_{\mathrm{SEG}} = \mathcal{L}_{\mathrm{BCE}}\big(\boldsymbol{H}_{\mathrm{prior}},\,\boldsymbol{M}_{\mathrm{gt}}\big) + \lambda_d\mathcal{L}_{\mathrm{DICE}}\big(\hat{\boldsymbol{M}},\,\boldsymbol{M}_{\mathrm{gt}}\big) + \lambda_f\mathcal{L}_{\mathrm{FOCAL}}\big(\hat{\boldsymbol{M}},\,\boldsymbol{M}_{\mathrm{gt}}\big). \qquad (8)$$

### 3.3 TRAINING AND INFERENCE

**Data Preparation.** Before entering the $\mathcal{F}_{\mathrm{enc}}$, we resize each image so that its longer side is 1024 pixels while preserving aspect ratio, then we pad it to $1024 \times 1024$. We apply the same transforms to the masks to maintain coordinate alignment during loss computation. For the policy $\pi_{\boldsymbol{\theta}}$, we cap the input at 705,600 pixels (900 vision tokens). If an image exceeds this cap, we downsample it while preserving aspect ratio for the policy input.

**Training Procedure.** As shown in Figure 2, during training we tokenize $(\boldsymbol{x}_{\mathrm{img}}, \boldsymbol{x}_{\mathrm{txt}})$, replicate each pair for $G$ times, and feed these copies to the $\pi_{\boldsymbol{\theta}}$ to generate $G$ responses. For each response in the group, the reward function assigns a scalar score, and the scores are converted into advantages for computing $\mathcal{L}_{\mathrm{GRPO}}$, which updates only the MLLM parameters. In the same batch, $\boldsymbol{x}_{\mathrm{img}}$ is resized and padded, then encoded by the vision backbone, and decoded to $\hat{\boldsymbol{M}}$ for computing $\mathcal{L}_{\mathrm{SEG}}$, which updates all trainable modules. We optimize both losses jointly in each iteration.

**Inference Procedure.** At inference, $(\boldsymbol{x}_{\mathrm{img}}, \boldsymbol{x}_{\mathrm{txt}})$ is used without replication. $\pi_{\boldsymbol{\theta}}$ runs with deterministic next token prediction to produce a single response that includes the concentration token. We then apply the same forward path as in training to produce mask logits. Finally, we remove padding, resize to the original image size, and threshold the logits at zero to obtain the binary mask.

## 4 EXPERIMENTS

**Research Questions.** In this section, we aim to answer the following research questions:

Table 1: Comparison of methods on RefCOCO, RefCOCO+, and RefCOCOg datasets.

| Model Type | Method | RefCOCO | | | RefCOCO+ | | | RefCOCOg | |
|---|---|---|---|---|---|---|---|---|---|
| | | val | testA | testB | val | testA | testB | val | test |
| Methods without LLMs | VLT | 67.5 | 70.5 | 65.2 | 56.3 | 61.0 | 50.1 | 55.0 | 57.7 |
| | CRIS | 70.5 | 73.2 | 66.1 | 62.3 | 68.1 | 53.7 | 59.9 | 60.4 |
| | LAVT | 72.7 | 75.8 | 68.8 | 62.1 | 68.4 | 55.1 | 61.2 | 62.1 |
| | ReLA | 73.8 | 76.5 | 70.2 | 66.0 | 71.0 | 57.7 | 65.0 | 66.0 |
| | X-Decoder | – | – | – | – | – | – | 64.6 | – |
| | SEEM | – | – | – | – | – | – | 65.7 | – |
| Latent Reasoning | LISA-7B | 74.9 | 79.1 | 72.3 | 65.1 | 70.8 | 58.1 | 67.9 | 70.6 |
| | LISA-13B | 76.0 | 78.8 | 72.9 | 65.0 | 70.2 | 58.1 | 69.5 | 70.5 |
| | PerceptionGPT-7B | 75.1 | 78.6 | 71.7 | 68.5 | 73.9 | 61.3 | 70.3 | 71.7 |
| | PerceptionGPT-13B | 75.3 | 79.1 | 72.1 | 68.9 | 74.0 | 61.9 | 70.7 | 71.9 |
| | PixelLM-7B | 73.0 | 76.5 | 68.2 | 66.3 | 71.7 | 58.3 | 69.3 | 70.5 |
| | LaSagnA-7B | 76.8 | 78.7 | 73.8 | 66.4 | 70.6 | 60.1 | 70.6 | 71.9 |
| | SegLLM-7B | 80.2 | 81.5 | 75.4 | 70.3 | 73.0 | 62.5 | 72.6 | 73.6 |
| | OMG-LLaVA-7B | 78.0 | 80.3 | 74.1 | 69.1 | 73.1 | 63.0 | 72.9 | 72.9 |
| | GroundHog-7B | 78.5 | 79.9 | 75.7 | 70.5 | 75.0 | 64.9 | 74.1 | 74.6 |
| | GLaMM-7B | 79.5 | 83.2 | 76.9 | 72.6 | 78.7 | 64.6 | 74.2 | 74.9 |
| | RAS-13B | 81.0 | 83.5 | 79.0 | 75.1 | 80.0 | **70.3** | 76.0 | 77.5 |
| Text-based Reasoning | SAM4MLLM-7B | 79.6 | 82.8 | 76.1 | 73.5 | 77.8 | 65.8 | 74.5 | 75.6 |
| | Seg-R1-3B | 69.9 | 76.0 | 64.9 | 59.1 | 66.8 | 50.9 | 67.9 | 67.3 |
| | Seg-R1-7B | 74.3 | 78.7 | 67.6 | 62.6 | 70.9 | 57.9 | 71.0 | 71.4 |
| | Seg-Zero-3B | – | 79.3 | – | – | 73.7 | – | – | 71.5 |
| | Seg-Zero-7B | – | 80.3 | – | – | 76.2 | – | – | 72.6 |
| | Text4Seg-7B | 79.3 | 81.9 | 76.2 | 72.1 | 77.6 | 66.1 | 72.1 | 73.9 |
| | Text4Seg-13B | 80.2 | 82.7 | 77.3 | 73.7 | 78.6 | 67.6 | 74.0 | 75.1 |
| Positional Prior | CoPRS-3B | 80.4 | 83.9 | 75.6 | 71.8 | 78.9 | 66.5 | 74.8 | 73.7 |
| | CoPRS-7B | **81.6** | **85.3** | **79.5** | **75.9** | **80.3** | 69.7 | **76.2** | 76.2 |

**RQ1:** Does CoPRS achieve higher accuracy in reasoning segmentation and state-of-the-art results on standard benchmarks compared to prior methods?

**RQ2:** How are the CoT, the positional prior $H_{\text{prior}}$, and the predicted mask $\hat{M}$ mutually correlated, i.e., does higher CoT quality align with stronger positional priors and better segmentation accuracy?

**RQ3:** Do the GRPO settings, supervised segmentation losses, and MLLM/vision backbone choices each contribute to performance, and does our unified objective with the default backbones outperform these alternatives?

## 4.1 EXPERIMENTAL SETUP

**Datasets and Metrics.** We evaluate CoPRS by conducting experiments on four datasets. We train CoPRS-3B and CoPRS-7B separately on the training sets of RefCOCO, RefCOCO+ and Ref-COCOg. To prevent data leakage, we remove from the training data all COCO images that appear in the validation or test splits of RefCOCO(+/g). We evaluate on the official validation and test splits of RefCOCO(+/g). We further assess zero-shot reasoning segmentation by evaluating on Reason-Seg (validation and test) without training on its images. Consistent with common practice in prior work (e.g., Lai et al. (2024)), we adopt intersection over union (IoU) metrics. Specifically, we report cIoU (the cumulative intersection over the cumulative union) on RefCOCO(+/g), and both cIoU and gIoU (mean of per-image IoU) on ReasonSeg.

**Baselines.** We compare our method with 20 prior works grouped into three categories. Methods without LLMs, including VLT (Ding et al., 2021), CRIS (Wang et al., 2022), LAVT (Yang et al., 2022), ReLA (Liu et al., 2023a), X-Decoder (Zou et al., 2023a), SEEM (Zou et al., 2023b), Grounded-SAM (Ren et al., 2024a), do not rely on LLM to encode instruction texts for generating masks. Latent reasoning methods, including LISA (Lai et al., 2024), PerceptionGPT (Pi et al., 2024), PixelLM (Ren et al., 2024b), LaSagnA (Wei et al., 2024), SegLLM (Wang et al., 2025a), OMG-LLaVA (Zhang et al., 2024a), GroundHog (Zhang et al., 2024b), GLaMM (Rasheed et al., 2024), RAS (Cao et al., 2025), take hidden features from a large language model and decode them into segmentation masks. Text-based reasoning methods, including SAM4MLLM (Chen et al., 2024), Seg-Zero (Liu et al., 2025), Seg-R1 (You & Wu, 2025), Text4Seg (Lan et al., 2025), use an MLLM to emit discrete location tokens—box/point coordinates or patch indices, and then convert them to

masks. For approaches available in multiple parameter scales, we report results for all the variants. RAS provides only a version with 13B parameters.

**Implementation Details.** We train on 8 NVIDIA A100 (80 GB) GPUs. Our implementation builds on the VERL codebase. Concretely, we weight the two components of reward function as 0.7 for mask and 0.3 for CoT format. Within the mask score, the coefficients for soft IoU, soft Dice, and hard IoU are set to 0.5, 0.2, and 0.3, respectively, and the format score is computed under specific regular expression rules for five conditions (see Section B.1). For GRPO, we use sampling numbers of 2, 4, and 8. Loss coefficients $\lambda_{\text{SEG}}$, $\lambda_d$ and $\lambda_f$ are set to 0.3, 3.0 and 10, respectively, for most batches. The base learning rate for the MLLM backbone is set to 2e-6; we apply multipliers of $25\times$ for the concentration query head, and $10\times/5\times$ for two submodules of mask decoder. We use the AdamW (Loshchilov & Hutter, 2019) optimizer with weight decay 0.01. We adopt OneCycleLR (Smith & Topin, 2019) as the learning rate scheduler, applying cosine decay to each parameter group down to one tenth of its peak learning rate. Full configurations are provided in Section B.3.

## 4.2 OVERALL PERFORMANCE (RQ1)

We compare CoPRS with prior state-of-the-art reasoning segmentation methods on two standard benchmarks: the RefCOCO series and ReasonSeg.

**Results on RefCOCO(+/g).** We follow standard evaluation protocols (Lai et al., 2024) and evaluate on the RefCOCO series. At matched model sizes, CoPRS-3B and CoPRS-7B achieve the best performance across all RefCOCO, RefCOCO+, and RefCOCOg splits (Table 1). Specifically, CoPRS-7B outperforms the latest reasoning methods on all the splits, trailing RAS-13B on only 2 of 8 splits. This advantage stems from our learning objectives, strengthening the CoT reasoning capability of CoPRS, which is crucial in reasoning segmentation.

Moreover, compared to Seg-R1 and Seg-Zero trained via GRPO, CoPRS achieves significant improvements at both model scales, with the 3B model surpassing their 7B counterparts. This fully demonstrates the effectiveness of our designed learnable concentration query in connecting reasoning and segmentation.

**Results on ReasonSeg.** We evaluate on ReasonSeg in a zero-shot setting to validate the generalization ability of CoPRS on complex reasoning segmentation scenarios. From Table 2, our CoPRS also demonstrates superior results on the complex reasoning segmentation task. Meanwhile, we find that methods trained with reinforcement learning, such as Seg-R1, Seg-Zero and our CoPRS, consistently outperform other methods, demonstrating the generalization benefits of reinforcement learning for segmentation models.

Table 2: Zero-shot comparison of methods on ReasonSeg dataset.

| Model Type | Method | val | | test | |
|---|---|---|---|---|---|
| | | gIoU | cIoU | gIoU | cIoU |
| Methods without LLMs | ReLA | 22.4 | 19.9 | 21.3 | 22.0 |
| | X-Decoder | 22.6 | 17.9 | 21.7 | 16.3 |
| | SEEM | 25.5 | 21.2 | 24.3 | 18.7 |
| | Grounded-SAM | 26.0 | 14.5 | 21.3 | 16.4 |
| Latent Reasoning | LISA-7B | 53.6 | 52.3 | 48.7 | 48.8 |
| | LISA-13B | 57.7 | 60.3 | 53.8 | 50.8 |
| | LaSagnA-7B | – | 47.2 | – | – |
| | SegLLM-7B | 57.2 | 54.3 | 52.4 | 48.4 |
| | GroundHog-7B | 56.2 | – | – | – |
| Text-based Reasoning | SAM4MLLM-7B | 46.7 | 48.1 | – | – |
| | Seg-R1-3B | 60.8 | 56.2 | 55.3 | 46.6 |
| | Seg-R1-7B | 58.6 | 41.2 | 56.7 | 53.7 |
| | Seg-Zero-3B | 58.2 | 53.1 | 56.1 | 48.6 |
| | Seg-Zero-7B | 62.6 | 62.0 | 57.5 | 52.0 |
| Positional Prior | CoPRS-3B | 61.3 | 60.6 | 57.8 | 52.7 |
| | CoPRS-7B | 65.2 | 64.5 | 59.8 | 55.1 |

## 4.3 CORRELATION ANALYSIS AND VISUALIZATION (RQ2)

**Correlation Analysis Methodology.** We first analyze the correlation between the positional prior $H_{\text{prior}}$ and the predicted mask $\hat{M}$ during both training and inference. We then analyze how the quality of CoT correlates with both $H_{\text{prior}}$ and $\hat{M}$, thereby linking the linguistic reasoning to the visual outputs. We plot the corresponding training losses and evaluation metrics as scatter points to make the relationship clear. Additionally, we use ordinary least square regression to plot the regression line $y = \hat{\alpha} + \hat{\beta}x$ and the mean confidence bands $\hat{y}(x) \pm \eta \, \text{s.e.} \, (\hat{y}(x))$, where s.e.$(\hat{y}) = \hat{\sigma}\sqrt{\frac{1}{n} + \frac{(x-\bar{x})^2}{\sum_i (x_i-\bar{x})^2}}$ with $\eta = 10$ for visual clarity and $\hat{\sigma}$ being residual standard error.

**Correlation between Heatmap and Mask.** During training, panels (a)–(d) in Figure 3 show blue points, each representing one training batch. The x-axis is $1 - \mathcal{L}_{\text{BCE}}(H_{\text{prior}}, M_{\text{gt}})$, which increases

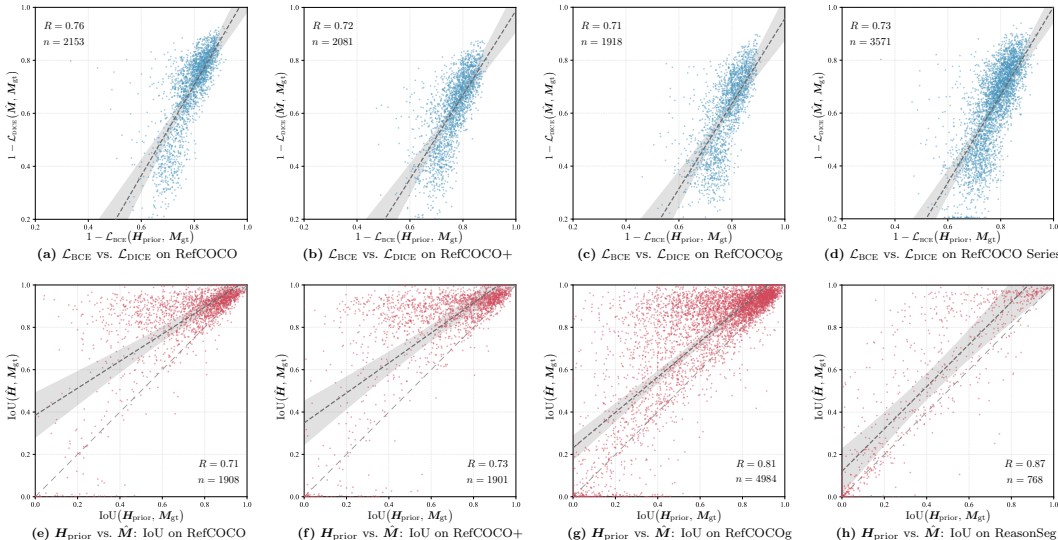

Figure 3: **Correlation analysis** between the positional prior $\boldsymbol{H}_{\text{prior}}$ and the predicted mask $\hat{\boldsymbol{M}}$ during training and inference on RefCOCO(+/g) and ReasonSeg. Each blue point represents one training batch, while each red point represents one inference instance. Ordinary least squares (OLS) regression lines and mean confidence bands are overlaid.

as the prior better matches $\boldsymbol{M}_{\text{gt}}$. The y-axis is $1 - \mathcal{L}_{\text{DICE}}\big(\hat{\boldsymbol{M}}, \boldsymbol{M}_{\text{gt}}\big)$, which is higher when $\hat{\boldsymbol{M}}$ converges to $\boldsymbol{M}_{\text{gt}}$. The points exhibit low dispersion, reflecting stable loss with batch size of 128. Across all datasets, the scatter patterns and correlation coefficients $R > 0.7$ indicate a strong positive association between $\boldsymbol{H}_{\text{prior}}$ and $\hat{\boldsymbol{M}}$.

During inference, panels (e)–(h) in Figure 3 show red points, each representing one inference instance. The x-axis is IoU between $\boldsymbol{H}_{\text{prior}}$ and $\boldsymbol{M}_{\text{gt}}$, i.e., the mask quality if the prior were used directly with no decoding. The y-axis is the IoU between $\hat{\boldsymbol{M}}$ and $\boldsymbol{M}_{\text{gt}}$, a standard segmentation metric. As in training, the scatter pattern and correlations $R > 0.7$ reveal a strong positive relationship across test splits. It is observed that the regression lines, confidence bands and most points lie above $y = x$. This trend indicates that the positional prior already concentrates well, while the decoder further refines it to a precise mask.

**Correlation between CoT and Segmentation Quality.** While Figure 3 already confirms the alignment between the heatmap and the final masks, it does not yet quantify how well the CoT reasoning itself aligns with these visual outputs. To make this link more explicit, we additionally use Gemini-2.5-Flash (Comanici et al., 2025) as an independent automatic evaluator. Inspired by Yin et al. (2025), we compute a consistency score in $[0, 1]$ (weighted average over four dimensions: logical correctness 0.3, task relevance 0.2, visual consistency 0.3, localization accuracy 0.2) between the image–instruction pair and the gen-

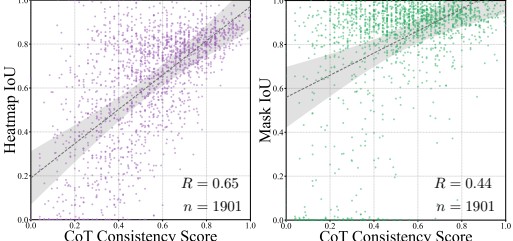

Figure 4: **Correlation** between CoT quality and segmentation quality (Heatmap/Mask IoU) on RefCOCO+. OLS results are overlaid.

erated CoT on the RefCOCO+ testA split. The scatter plots in Figure 4 show a clear positive correlation between CoT consistency scores and both Heatmap IoU and Mask IoU. This quantitative evidence directly supports that better CoT reasoning quality leads to better segmentation performance in CoPRS.

**Visualization Results.** We present zero-shot visualizations on ReasonSeg, as shown in Figure 5. After MCoT reasoning, the positional prior indicates all instances relevant to the instruction (yellow), with the target instance most strongly concentrated (deep red). Figure 8 in Appendix presents

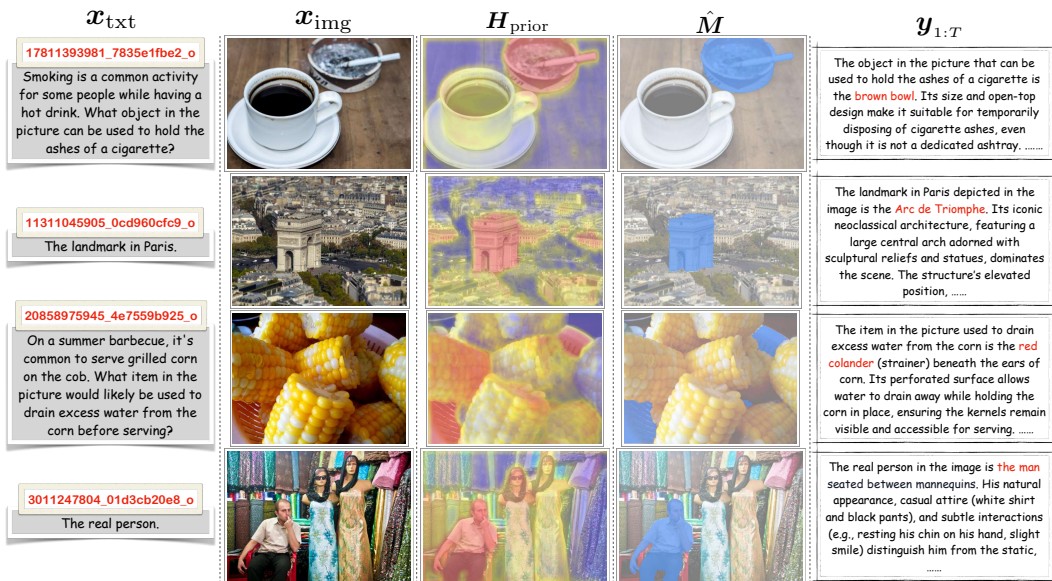

Figure 5: **Sample visualizations.** With sample ID exposed, all samples are from the ReasonSeg test split. From left to right: image-text pair, positional prior, predicted mask, and chain of thought.

additional visualizations. Additional failure cases in Figure 7 in the Appendix show that CoPRS mainly struggles with very small objects that disappear at our current input resolution, and dense groups of similar instances where text alone cannot reliably disambiguate the target.

## 4.4 Ablation Study (RQ3)

To gain a deeper understanding of the contributing factors, we perform ablation studies on Ref-COCO+ with different MLLM backbones and varied vision backbones, and further ablations of CoPRS-7B on RefCOCO+, RefCOCOg, and ReasonSeg. We systematically examine MLLM backbone choice, vision backbone choice, GRPO group size, training mode, reward coefficients, and segmentation loss combinations.

**MLLM Backbone.** For ablating the MLLM backbone, we additionally train CoPRS with LLaVA-1.5-7B/13B on RefCOCO+. Table 3 reports cIoU metrics of CoPRS versions with both LLaVA-1.5 and Qwen2.5-VL series. As expected, performance increases with backbone capacity, but the gains across different MLLM backbones are relatively modest. This indicates that CoPRS is not sensitive to the specific MLLM

Table 3: Effect of MLLM Backbone Choice. **Gray row** denotes the default backbone.

| Method | Backbone | val | testA | testB |
|---|---|---|---|---|
| CoPRS-3B | Qwen2.5-VL | 71.8 | 78.9 | 66.5 |
| CoPRS-7B | Qwen2.5-VL | **75.9** | **80.3** | 69.7 |
| CoPRS-7B | LLaVA-1.5 | 73.1 | 79.0 | 66.4 |
| CoPRS-13B | LLaVA-1.5 | 75.5 | **80.3** | **70.7** |

architecture and that our improvements largely transfer across different backbone choices. Together with the comparisons to prior work under the same LLaVA-1.5 backbone (Table 1), this suggests that our gains are complementary to backbone strength rather than being tied to a particular MLLM.

**Vision Backbone.** As shown in Table 4, we ablate SAM backbones (ViT-B/L/H) on RefCOCO+ with a fixed Qwen2.5-VL-7B MLLM and report the total parameters of the full pipeline. Larger vision backbones bring slightly better segmentation performance, but the improvement is modest and the overall trend remains stable across sizes.

Table 4: Effect of Vision Backbone Choice. **Gray row** denotes the default backbone.

| Backbone | #Params(B) | val | testA | testB |
|---|---|---|---|---|
| ViT-B | 8.38 | 73.2 | 77.3 | 67.0 |
| ViT-L | 8.60 | 74.8 | 78.9 | 68.5 |
| ViT-H | 8.93 | **75.9** | **80.3** | **69.7** |

Additionally, vision backbones constitute only a small portion of the total parameters, so scaling them up only marginally increases overall computational cost.

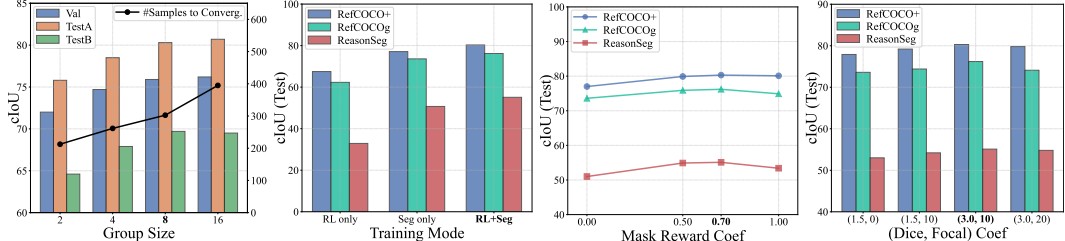

(a) Effect of GRPO group size

(b) Effect of training mode choice

(c) Effect of mask reward coefficient.

(d) Effect of segmentation loss coefficients.

Figure 6: **Ablation studies** on GRPO group size, training mode, mask reward coefficient, and segmentation loss coefficients. (a) is evaluated on all splits of RefCOCO+, while (b)–(d) are evaluated on the test split of each dataset. **Bold x-axis labels** mark the default settings.

**GRPO Group Size.** We study the effects of GRPO group size during training. The group size $G$ denotes the number of responses sampled per question during rollout. As shown in Figure 6a, increasing $G$ improves performance across splits of RefCOCO+. To quantify efficiency, we also report the total number of GRPO samples required to reach convergence (loss fluctuation $< 10\%$ over 300 steps) for $G \in \{2, 4, 8, 16\}$. Particularly, the number of samples for convergence does not grow linearly with $G$, because larger groups offer more diverse candidates per step, improving exploration and the contrast between positive and negative samples. Empirically, we find that $G = 8$ strikes a good trade-off between efficiency and performance.

**Training Modes.** We compare reinforcement learning, segmentation supervision, and a combined objective for CoPRS-7B. As shown in Figure 6b, the combined objective achieves the best performance. This suggests that reinforcement learning strengthens reasoning, while supervised signals sharpen mask generation. Together they are more effective for complex reasoning segmentation.

**Reward Coefficients.** We evaluate the impact of reward mixing ratio between mask reward score and format score. Figure 6c compares their combinations, where the format score is one minus the mask score. As the coefficient on the mask reward increases from 0 to 0.7, cIoU improves across all three datasets, but pushing it further to 1.0 slightly degrades performance. This pattern suggests that the segmentation term is the main driver of segmentation quality, while keeping a small contribution from the format score helps regularize the policy and improves generalization, especially on out of distribution data (ReasonSeg). We set the 0.7/0.3 weighting by default, with the segmentation reward dominant and the format score acting as a regularizer, and Figure 6c supports this choice.

**Segmentation Loss Combinations.** We compare segmentation loss configurations with varying coefficients (see Figure 6d) to assess the contribution of each component, with BCE weight fixed at 1. To avoid the prohibitive cost of LLM experiments, we only probe a few representative weight settings, which already show trends consistent with our expectations. Adding a focal loss term, which emphasizes hard pixels and fine-grained structures, improves segmentation performance. The relative weight between focal and dice loss also affects the balance between global and local mask quality.

## 5 CONCLUSIONS

In this work, we propose CoPRS, connecting language reasoning with segmentation via an interpretable and differentiable interface. CoPRS implements this idea with a learnable concentration query to produce a positional prior instantiated as a heatmap, from which precise masks are decoded, within a unified framework combining reinforcement learning and segmentation supervision. This interface avoids feeding hidden features to the decoder or representing positions in text, instead providing a direct, interpretable alignment between reasoning and mask generation. Empirically, CoPRS attains strong performance across datasets. Further analysis shows that CoT trajectory and heatmap quality strongly correlate with final mask accuracy, and sample visualizations show the same pattern. Overall, CoPRS delivers strong concentration from reasoning and predicts precise masks in a unified formulation, providing a starting point for perception aligned with instructions.

## REPRODUCIBILITY STATEMENT

**Reproducibility Statement.** We point readers to the fundamental setup in Experimental Setup (Section 4.1), and to the appendix Implementation Details (Section B), which concisely summarizes the pipeline implementation (Section B.1), the design details (Section B.2) and the training configuration (Section B.3). These sections contain the information needed to reproduce our results.

**LLM Usage Statement.** Consistent with policies on LLM usage, we used an LLM only for language polishing (see Section B.4 for details). All ideas, experiments, and analyses were produced and verified by the authors, who take full responsibility.

## ACKNOWLEDGMENTS

We sincerely thank the anonymous reviewers and chairs for their efforts and suggestions, greatly helping us improve the manuscript. This work is supported in part by the National Natural Science Foundation of China under grants 62536003 and 624B2088, and in part by the project of Peng Cheng Laboratory (PCL2025A14).

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

# A APPENDIX: GRPO THEORY AND ADDITIONAL RELATED WORK

## A.1 GROUP RELATIVE POLICY OPTIMIZATION

The reasoning ability of MLLMs is a key factor that influences the reasoning segmentation performance. Since Reinforcement Learning (RL) is an effective way to improve the reasoning ability of LLMs and MLLMs, we employ it to enhance the reasoning segmentation capability of our method.

Proximal Policy Optimization (PPO) (Schulman et al., 2017) is widely used in the RL fine-tuning stage of LLMs. PPO is an actor-critic RL algorithm, which optimizes LLMs by maximizing the following surrogate objective:

$$\mathcal{L}_{\text{PPO}} = \mathbb{E}\left[q \sim P(Q), o \sim \pi_{\boldsymbol{\theta}_{\text{old}}}(O|q)\right] \frac{1}{|o|} \sum_{t=1}^{|o|} \min\left[\frac{\pi_{\boldsymbol{\theta}}(o_t|q,\boldsymbol{o}_{<t})}{\pi_{\boldsymbol{\theta}_{\text{old}}}(o_t|q,\boldsymbol{o}_{<t})} A_t, \text{clip}\left(\frac{\pi_{\boldsymbol{\theta}}(o_t|q,\boldsymbol{o}_{<t})}{\pi_{\boldsymbol{\theta}_{\text{old}}}(o_t|q,\boldsymbol{o}_{<t})}, 1-\varepsilon, 1+\varepsilon\right) A_t\right] \quad (9)$$

where $\pi_{\boldsymbol{\theta}}$ and $\pi_{\boldsymbol{\theta}_{old}}$ are the current and old policy models, and $q, o$ are questions and outputs sampled from the question dataset and the old policy $\pi_{\boldsymbol{\theta}_{old}}$, respectively. $\varepsilon$ is a clipping-related hyper-parameter introduced in PPO for stabilizing training. The advantage, $A_t$, is based on the reward $\{r_{\geq t}\}$ and a learned value function $V_\psi$, computed by applying Generalized Advantage Estimation (GAE) (Schulman et al., 2015). Furthermore, a per-token KL penalty from a reference model is added to the reward at each token to mitigate over-optimization of the reward model (Ouyang et al., 2022), denoted as:

$$r_t = r_\varphi(q, \boldsymbol{o}_{\leq t}) - \beta \log \frac{\pi_{\boldsymbol{\theta}}(o_t|q, \boldsymbol{o}_{<t})}{\pi_{\text{ref}}(o_t|q, \boldsymbol{o}_{<t})} \quad (10)$$

where $r_\varphi$ is the reward model, $\pi_{\text{ref}}$ is the reference model, which is usually the initial policy model, and $\beta$ is the coefficient of the KL penalty.

PPO relies on a separate value function that is typically another model of comparable size to the policy model, imposing heavy memory and computational costs. Additionally, the value function is treated as a baseline in the calculation of the advantage for variance reduction. Moreover, in the LLM context, usually only the last token is assigned a reward score by the reward model, which may complicate the training of a value function that is accurate at each token. Group Relative Policy Optimization (GRPO) (Shao et al., 2024) is proposed to address these drawbacks by obviating the need for additional value function approximation as in PPO, and using the average reward of multiple sampled outputs, produced in response to the same question, as the baseline. Specifically, for each question $q$, GRPO samples a group of outputs $\{o_1, o_2, \cdots, o_G\}$ from the old policy $\pi_{\boldsymbol{\theta}_{old}}$ and then optimizes the policy model by maximizing the following objective:

$$\mathcal{L}_{\text{GRPO}} = \mathbb{E}_{q \sim P(Q), \{o_i\}_{i=1}^G \sim \pi_{\boldsymbol{\theta}_{\text{old}}}(O|q)}$$
$$\left\{\frac{1}{G}\sum_{i=1}^{G}\frac{1}{|o_i|}\sum_{t=1}^{|o_i|}\min\left[\frac{\pi_{\boldsymbol{\theta}}(o_{i,t}|q,\boldsymbol{o}_{i,<t})}{\pi_{\boldsymbol{\theta}_{\text{old}}}(o_{i,t}|q,\boldsymbol{o}_{i,<t})}\hat{A}_{i,t}, \text{clip}\left(\frac{\pi_{\boldsymbol{\theta}}(o_{i,t}|q,\boldsymbol{o}_{i,<t})}{\pi_{\boldsymbol{\theta}_{\text{old}}}(o_{i,t}|q,\boldsymbol{o}_{i,<t})}, 1-\varepsilon, 1+\varepsilon\right)\hat{A}_{i,t}\right] - \beta\mathbb{D}_{\text{KL}}\left[\pi_{\boldsymbol{\theta}}\|\pi_{\text{ref}}\right]\right\} \quad (11)$$

where $\varepsilon$ and $\beta$ are hyper-parameters, and $\hat{A}_{i,t}$ is the advantage calculated based on relative rewards of the outputs inside each group only. For each question $q$, a group of outputs $\{o_1, o_2, \cdots, o_G\}$ are sampled from the old policy model $\pi_{\boldsymbol{\theta}_{old}}$. The score of the outputs is obtained through a reward model, yielding $G$ rewards $\{r_1, r_2, \cdots, r_G\}$ correspondingly. The advantages $\hat{A}_{i,t}$ for all tokens in an output are defined as the normalized reward, i.e., $\hat{A}_{i,t} = \tilde{r}_i = \frac{r_i - \text{mean}(\boldsymbol{r})}{\text{std}(\boldsymbol{r})}$. In addition, GRPO directly adds the KL divergence between the trained policy and the reference policy to the loss, avoiding complicating the calculation of $\hat{A}_{i,t}$. The KL divergence is estimated by the following unbiased estimator:

$$\mathbb{D}_{\text{KL}}\left[\pi_{\boldsymbol{\theta}}\|\pi_{\text{ref}}\right] = \frac{\pi_{\text{ref}}(o_{i,t}|q,\boldsymbol{o}_{i,<t})}{\pi_{\boldsymbol{\theta}}(o_{i,t}|q,\boldsymbol{o}_{i,<t})} - \log\frac{\pi_{\text{ref}}(o_{i,t}|q,\boldsymbol{o}_{i,<t})}{\pi_{\boldsymbol{\theta}}(o_{i,t}|q,\boldsymbol{o}_{i,<t})} - 1 \quad (12)$$

## A.2 ADDITIONAL RELATED WORK

**GRPO Guided Reinforcement Learning.** The GRPO (Shao et al., 2024) strategy addresses reward hacking in RLHF (Dong et al., 2024) by penalizing deviation from a reference policy. However, its reliance on a static reference limits adaptability. This spurred key optimizations: Dynamic Advantage-based Policy Optimization (DAPO) (Yu et al., 2025b) introduces a moving trust region by dynamically updating the reference policy via an exponential moving average, enabling more stable, long-term improvement. Another significant limitation of the original GRPO is its token-level

optimization, which can be computationally intensive and may lead to training instability. Addressing this, Sequence-wise Policy Optimization (GSPO) (Zheng et al., 2025) was proposed to shift the optimization granularity from the token level to the sequence level. By defining a sequence-level importance ratio and advantage, GSPO significantly reduces computational overhead and improves training stability, especially for large-scale models.

**Multimodal Chain-of-Thought.** Multimodal chain-of-thought (MCoT) (Wang et al., 2025b) reasoning has recently attracted substantial attention, particularly in its integration with MLLMs. Early implementations, such as Multimodal-CoT (Zhang et al., 2024c), have established a basic MCoT pattern by generating intermediate rationales before predictions. MC-CoT (Tan et al., 2024) further refines this paradigm by employing word-level majority during training to enhance the quality of generated rationales. The dependence on high-quality MCoT training data hinders the further improvement of the inference ability of traditional methods. Most recently, the great success of Deepseek-R1 (Guo et al., 2025) has provided a way (i.e., GRPO) to enhance LLM inference capabilities through model autonomous exploration without the need for expensive CoT annotation data. Inspired by this, subsequent works utilize the GRPO strategy to efficiently enhance the reasoning ability of MLLMs. For example, Vision-R1 (Huang et al., 2025) first utilizes existing MLLM and DeepSeek-R1, as well as data filtering, through modal bridging to generate multimodal cold start CoT data, and then applies GRPO to further enhance the model's inference capability. Perception-R1 (Yu et al., 2025a) explores the effects of RL on different perception tasks and optimizes the reward modeling to support perception policy learning. In addition, Chain-of-Shot (Hu et al., 2025) further extends GRPO strategy to optimize frame sampling via binary video summaries. In this work, we study a heatmap-based positional prior that couples MCoT with precise positional perception in a unified training framework for GRPO strategy and segmentation supervision, addressing the gap between high-level reasoning and pixel-level segmentation.

# B  APPENDIX: IMPLEMENTATION DETAILS

## B.1  PIPELINE IMPLEMENTATION

We build on the VERL codebase, which was originally designed for PPO and extended with GRPO functionality.

**Sharding Strategy.** We shard the VLLM/policy component using Fully Sharded Data Parallel (FSDP), partitioning parameters across devices during training. The lightweight segmentation modules (query head, Q–V attention, and mask decoder) are left unsharded to avoid FSDP overhead and keep their compute/memory costs low. We apply tensor parallelism across attention heads during autoregressive decoding.

**FSDP Workers.** We precompute image features offline to reduce compute, so the frozen vision backbone is excluded from the training loop. Our framework uses three FSDP workers. (i) The **actor** contains all trainable modules (the MLLM and the segmentation components) and is responsible for parameter updates. (ii) The **rollout worker** runs the MLLM only, taking image and text inputs to generate responses via next token prediction. (iii) The frozen **reference worker** runs an MLLM as the reference policy to compute the KL term in $\mathcal{L}_{\text{GRPO}}$ (eq. (7)) and includes the segmentation modules to decode masks used for computation of mask reward scores and group advantages.

**Training Pipeline Implementation.** For each annotation, the rollout worker generates $G$ responses for the image–text pair with the current policy by next token prediction, caching the tokens and their log probabilities. The frozen reference worker then runs forward without gradients on the same inputs to compute reference log probabilities for those sampled responses and to decode a mask used in the mask based reward. From each response and its mask signal we compute a scalar reward and convert rewards to group advantages. Next, the actor worker runs forward to obtain the policy log probabilities for the sampled responses and the predicted mask. We form the GRPO objective from the actor log probabilities, the stored old log probabilities from rollout, the reference log probabilities, and the advantages, and we form the segmentation objective from the predicted mask and the ground truth mask. The two objectives are summed and optimized jointly in a single backward pass, updating all trainable modules.

---

**Algorithm 1** Generation of positional prior $\boldsymbol{H}_{\text{prior}}$

---

**Require:** Image $\boldsymbol{x}_{\text{img}}$; concentration token embedding $\boldsymbol{e}_{\text{conc}}$; image encoder $\mathcal{F}_{\text{enc}}$; query head $\mathcal{F}_{\text{head}}$;
    fusion network $\mathcal{F}_{\text{fuse}}$; projection matrices $\{\boldsymbol{W}_i^Q, \boldsymbol{W}_i^K\}_{i=1}^{n_{\text{head}}}$
**Ensure:** Positional prior $\boldsymbol{H}_{\text{prior}} \in \mathbb{R}^{H \times W}$

1:   $\boldsymbol{K} \leftarrow \mathcal{F}_{\text{enc}}(\boldsymbol{x}_{\text{img}})$                                       $\triangleright$ $\boldsymbol{K} \in \mathbb{R}^{H \times W \times d_k}$
2:   $\boldsymbol{Q} \leftarrow \mathcal{F}_{\text{head}}(\boldsymbol{e}_{\text{conc}})$                                        $\triangleright$ $\boldsymbol{Q} \in \mathbb{R}^{d_q}$
3:   **for** $i = 1$ to $n_{\text{head}}$ **do**
4:       $\boldsymbol{K}_i \leftarrow \boldsymbol{K}\boldsymbol{W}_i^K$                            $\triangleright$ $\boldsymbol{K}_i \in \mathbb{R}^{H \times W \times d_h}$
5:       $\boldsymbol{q}_i \leftarrow \boldsymbol{Q}\boldsymbol{W}_i^Q$                              $\triangleright$ $\boldsymbol{q}_i \in \mathbb{R}^{d_h}$
6:       **for** $(u, v) \in \{1, \dots, H\} \times \{1, \dots, W\}$ **do**
7:           $S_i(u, v) \leftarrow \dfrac{1}{\sqrt{d_h}}\, \boldsymbol{q}_i^\top \boldsymbol{K}_i(u, v)$           $\triangleright$ $S_i(u, v) \in \mathbb{R}$
8:       **end for**
9:   **end for**
10: $\boldsymbol{H}_{\text{prior}} \leftarrow \mathcal{F}_{\text{fuse}}\big( [\boldsymbol{S}_i]_{i=1}^{n_{\text{head}}} \big)$    $\triangleright$ $\mathcal{F}_{\text{fuse}}$: small conv fusion head, $\mathbb{R}^{n_{\text{head}} \times H \times W} \to \mathbb{R}^{H \times W}$
11: **return** $\boldsymbol{H}_{\text{prior}}$

---

## B.2 Design Details

**Reward Function Design.** We use a scalar mask score in $[0, 1]$: given predicted mask and ground truth mask, we compute three overlap metrics (soft IoU, soft Dice, and hard IoU) and take their weighted sum with fixed coefficients 0.5, 0.2, and 0.3, respectively, providing a stable localization signal for how well the prediction covers the instance. For valid outputs, the score is 1.0 by default and is reduced to 0.9 if the <think> content is longer than 2048 characters, or if any non-whitespace text appears before <think> or after the special token. Thus the five canonical cases are: invalid (0.0); valid and clean (1.0); valid but long <think> (0.9); valid but extra text before <think> (0.9); valid but extra text after the special token (0.9). For each sample, we take a weighted sum of these two components as the final reward that is assigned to the last valid response token so that GRPO updates the entire trajectory. The relative weights are specified in Section 4.4.

**Positional Prior Heatmap Generation.** To make the computation of the positional prior $\boldsymbol{H}_{\text{prior}}$ fully reproducible, we detail the heatmap generation procedure in Algorithm 1, starting from the image keys $\boldsymbol{K}$, the concentration query $\boldsymbol{Q}$, and the per-head scaled dot-product scores $S_i(u, v)$. The convolutional fusion head $\mathcal{F}_{\text{fuse}}$ then aggregates $\{\boldsymbol{S}_i\}_{i=1}^{n_{\text{head}}}$ into the final positional prior $\boldsymbol{H}_{\text{prior}} \in \mathbb{R}^{H \times W}$.

## B.3 Training Configuration

Data and preprocessing. We train on the RefCOCO series. The maximum prompt length is 1300 tokens and the maximum response length is 2000 tokens. For the policy input, images are capped at 705,600 pixels and downsampled if needed; a minimum of 3,136 pixels is enforced. SAM ViT-H features initialize the vision branch.

Hardware and precision. Experiments run on a single node with 8 GPUs. Computation uses bfloat16 for model parameters and fp32 for reductions and buffers.

Parallelism. The policy (VLLM) is trained with Fully Sharded Data Parallel. The rollout service uses tensor parallelism of size 4. The reference worker is also sharded; optimizer state is offloaded.

Batching. Global batch size is 16 (before repeating $G$ times for GRPO). For the actor, micro-batch per device is 2 for updates and 8 for experience collection. Rollout batch size is 16 and the group size is $G = 8$ responses per input.

Optimization. We use AdamW with weight decay 0.01 and $(\beta_1, \beta_2) = (0.9, 0.999)$. The base learning rate is $1.6 \times 10^{-6}$ with multipliers $25\times$ (query head), $10\times$ (position/prompt encoder), and $5\times$ (mask decoder). Gradient clipping uses a max norm of 1.0. The schedule is one cycle with a final division factor of about 6.7 and no warmup. Total planned training steps are 31,250. Gradient checkpointing is enabled.

GRPO settings. We use GRPO with sampling number 8, clip ratio 0.2, group-relative advantages, and a fixed KL penalty coefficient 0.2 (low-variance form). The entropy coefficient is 0.0.

Segmentation objectives. Unless noted, $\lambda_{\text{SEG}} = 0.3$, $\lambda_d = 1.5$, and $\lambda_f = 0.0$ at the start; at step 1,500 we set $\lambda_d = 3.0$ and $\lambda_f = 10.0$. Losses are computed only on the valid (unpadded) region.

Rollout and decoding. Rollouts use a VLLM backend with sampling enabled (temperature 1.0, top-p 1.0, top-k disabled). Execution uses bfloat16, up to 64 concurrent sequences, and a cap of 17,408 batched tokens. Chunked prefill is enabled. One image is used per sample.

### B.4 LLM USAGE STATEMENT

In preparing this paper, we used a large language model (LLM) for polishing at the sentence level. We do not directly include the text generated by LLM in our paper. Instead, we use it solely as a reference and for guidance. The model was given the following prompt to guide the text refinement process:

"Slightly polish it sentence by sentence, and give the reasons. Not latex code. Disable online search and do not find citations yourself. You must avoid changing any statistics and avoid distorting my statements."

This prompt was specifically designed to ensure that the LLM's revisions were limited to language refinement and that no statistics or experimental results were altered. The LLM was also instructed not to perform any online searches or generate citations. All final content, including experimental data and results, remains the responsibility of the authors.

## C  APPENDIX: ADDITIONAL SAMPLE VISUALIZATIONS

**Successful Cases.** In Figure 8, we present instances from the same category and those relevant to the instruction, all showing elevated responses in the heatmap (yellow regions). More importantly, the heatmap concentrates on the instance specified by the instruction, producing a sharp peak over the target (deep red regions). This concentration guides the decoder, yielding masks with accurate boundaries. These results indicate that the MLLM reasons over the image and text input and identifies the correct referent, while the positional prior concentrates the instances for further precise mask prediction.

**Failure Cases.** In Figure 7, the first two rows depict scenes with many nearby instances, while the last three rows contain very small targets. Two failure modes emerge. (i) Resolution bottleneck: the positional prior is computed at 256×256 and the SAM embeddings at 64×64; when the longer image side exceeds 2k pixels, tiny objects can vanish after resizing and the decoder cannot reliably recover them. (ii) Same class crowd ambiguity: in dense groups of similar objects (e.g., crowds of people), the positional prior often spreads across many candidates with weak contrast, suggesting that a text only instruction is insufficient to disambiguate near duplicates and that the model has not fully learned the subtle semantic cues needed. These observations suggest that higher resolution inputs or multi-scale features, together with stronger instance level language grounding, are likely to improve performance on such cases.

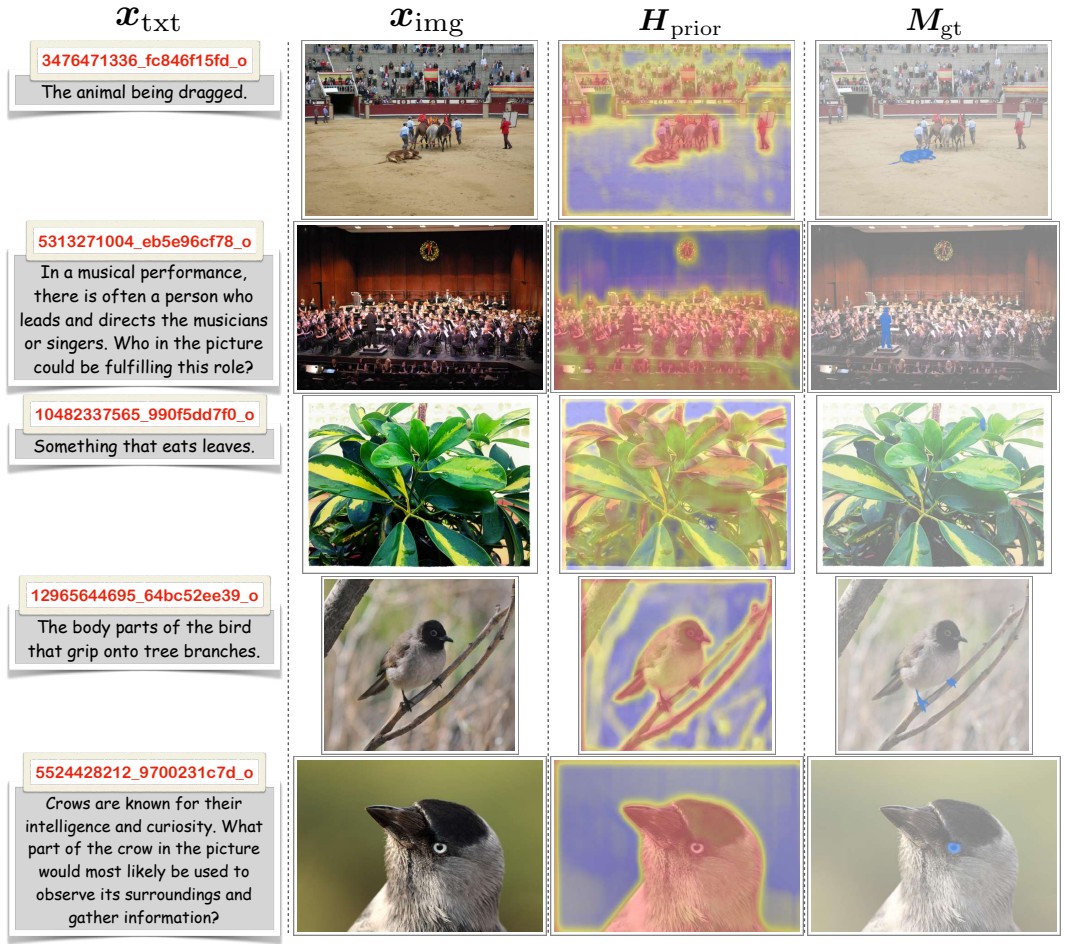

Figure 7: **Failure cases.** With sample ID exposed, all samples are from the ReasonSeg test split. From left to right: image-text pair, positional prior and ground truth mask.

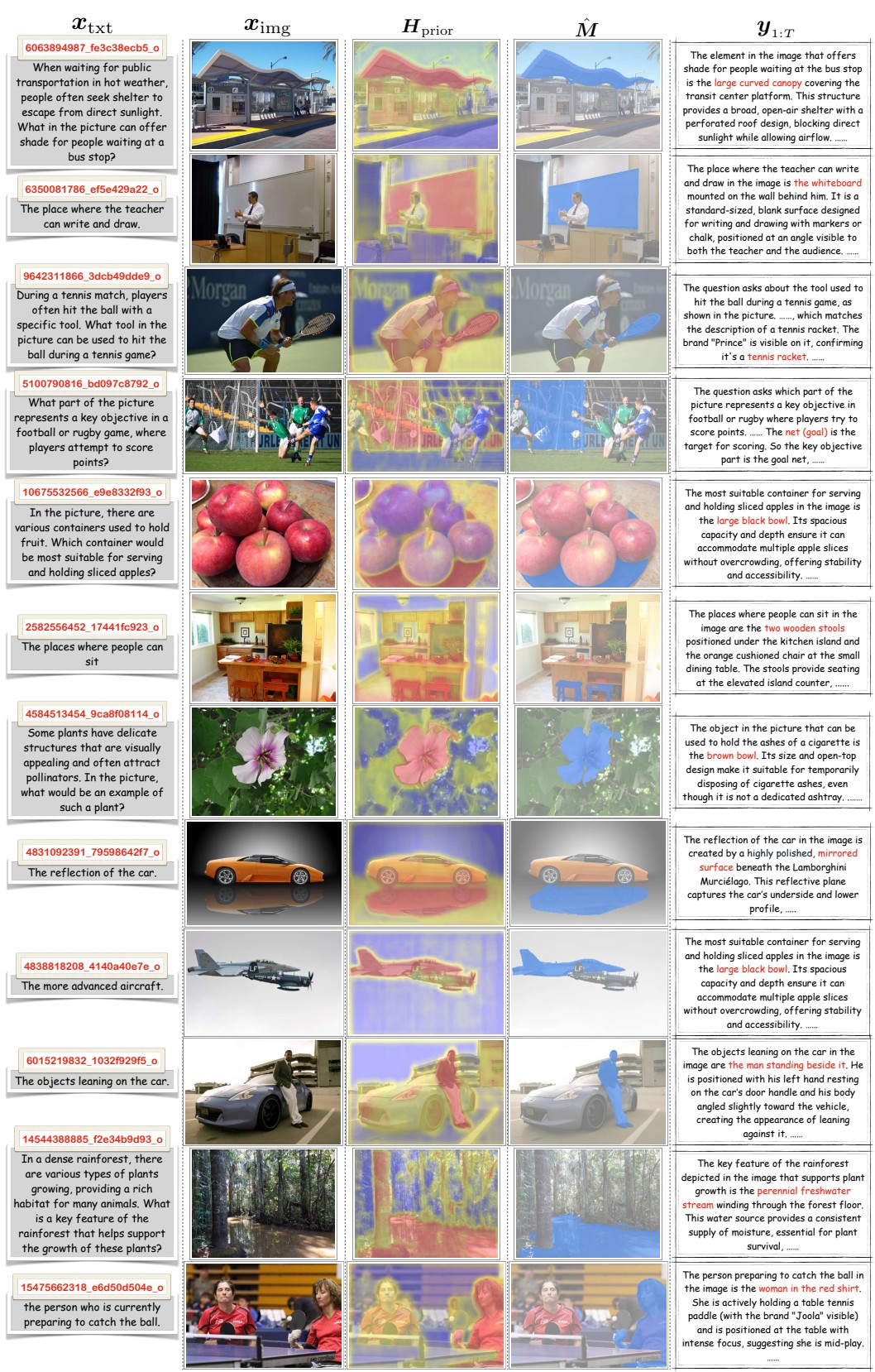

Figure 8: **Additional successful cases.** With sample ID exposed, all samples are from the Reason-Seg test split. From left to right: image-text pair, positional prior, predicted mask, and response.

