# OpenReview forum: "CoPRS: Learning Positional Prior from Chain-of-Thought for Reasoning Segmentation"
_ICLR.cc/2026/Conference — ICLR 2026 Poster_

### Official Review · Reviewer_XDcj · 2025-10-30

**Soundness:** 2
**Presentation:** 3
**Contribution:** 2
**Rating:** 6
**Confidence:** 3

**Summary:**

This paper introduces CoPRS, a multimodal chain-of-thought (MCoT)-based method for reasoning segmentation that bridges language reasoning to pixel-level segmentation via a differentiable positional prior (heatmap). The key contributions include:

- A novel interface using a learnable "concentration token" to generate positional priors from MCoT reasoning, providing interpretable intermediate heatmaps.
- A unified training framework combining Group Relative Policy Optimization (GRPO) for language reasoning and segmentation supervision for mask refinement.
- State-of-the-art performance on RefCOCO series and ReasonSeg benchmarks, with ablation studies demonstrating the correlation between heatmap quality and segmentation accuracy.

**Strengths:**

*Originality*: The integration of MCoT reasoning with dense positional priors offers a fresh perspective compared to existing latent-feature or text-coordinate paradigms. The use of GRPO for joint language-reasoning segmentation optimization is a creative combination of reinforcement learning and segmentation tasks.
*Clarity*: The architecture diagram (Figure 2) effectively illustrates the pipeline. The distinction between training modes (GRPO vs. supervised) is well articulated.
*Significance*: Addresses a critical gap in interpretability for reasoning segmentation systems. The reported 3B/7B model improvements over text-coordinate baselines (e.g., +4.7 cIoU over Seg-R1 on RefCOCOg) demonstrate practical value.

**Weaknesses:**

*Limited Baseline Comparison*: While compared to latent-reasoning and text-based methods, there is no direct comparison with recent hybrid approaches like PerceptionGPT-R1 or RAS-13B under identical training protocols. This leaves uncertainty about absolute performance claims.
*Ablation Depth*: The GRPO hyperparameter study (Table 4) only tests sampling numbers {2,4,8} without justifying this range. More analysis is needed on how group size affects exploration-exploitation tradeoffs in segmentation contexts.
*Methodological Ambiguity*: The heatmap generation process (Eq. 3-4) lacks critical implementation details, e.g., how the MLP maps vision backbone outputs to keys, or why two convolutional layers were chosen for F<sup>fuse</sup>. This hinders reproducibility.
*Theoretical Limitations*: While empirical correlations are shown, there is no formal analysis of why GRPO’s group-relative advantages are particularly suited for segmentation tasks compared to standard PPO.

**Questions:**

**Q1**: How does CoPRS handle ambiguous positional priors in crowded scenes (Figure 5)? The failure cases suggest sensitivity to instance density – could multi-scale features or contrastive learning between instances mitigate this?
**Q2**: For GRPO sampling numbers (Table 4), what computational overhead does G=8 introduce compared to G=2? A latency/accuracy tradeoff analysis would help practitioners.
**Q3**: The correlation analysis (Figure 3) shows association but not causation between heatmaps and masks. Could you disentangle whether performance gains stem primarily from GRPO-enhanced reasoning or the decoder architecture?
**Suggestion 1**: Add comparisons with RAS-13B and PerceptionGPT-R1 using comparable model sizes.
**Suggestion 2**: Include an ablation on the vision backbone (e.g., ViT-H vs. ViT-L) to clarify performance dependencies.
**Suggestion 3**: Provide pseudocode or extended equations for the heatmap generation process (Section 3.1) to improve reproducibility.

**Rebuttal Potential**: Addressing the baseline comparison gap (Suggestion 1) and providing theoretical justification for GRPO’s effectiveness in segmentation could significantly strengthen the contribution narrative. Clarifying the heatmap implementation details (Suggestion 3) would enhance methodological rigor.

---

> ### Author Response · Authors · 2025-11-25
> **Response to Reviewer XDcj (1/3)**
>
> We sincerely thank you for the insightful feedback. We are encouraged by your recognition of the work’s **originality** in integrating MCoT with dense positional priors, as well as the **practical significance** of our performance improvements over text-coordinate baselines. In the following section, we address your specific concerns and questions point-by-point.
>
> ### XDcj-W1
> > Limited Baseline Comparison: While compared to latent-reasoning and text-based methods, there is no direct comparison with recent hybrid approaches like PerceptionGPT-R1 or RAS-13B under identical training protocols. This leaves uncertainty about absolute performance claims.
>
> Your concern about the choice of MLLM backbone is very reasonable.
>
> RAS-13B is built on LLaVA-13B, and PerceptionGPT also provides both 7B and 13B variants. To reduce this ambiguity, we ran **additional experiments** on RefCOCO+. We train CoPRS with LLaVA-1.5 backbones of 7B and 13B, and compare the results with recent methods that use the same LLaVA-1.5 backbone (**See the table below**), including RAS-13B and PerceptionGPT-R1. Performance increases with backbone size, but in both the 7B and 13B settings CoPRS **consistently achieves higher scores** than methods built on the same backbone, suggesting that our improvements go beyond backbone scaling.
>
> | Method               | LLaVA-1.5 Size | Year  | Val cIoU | TestA cIoU | TestB cIoU |
> |:---------------------|:--------------:|:-----:|:--------:|:----------:|:----------:|
> | SegLLM               |      7B        | 2025  | 70.3     | 73.0       | 62.5       |
> | UniRES               |      7B        | 2025  | 71.6     | 76.0       | 64.4       |
> | PerceptionGPT        |      7B        | 2025  | 68.5     | 73.9       | 61.3       |
> | Text4Seg             |      7B        | 2025  | 72.1     | 77.6       | 66.1       |
> | CoPRS (Ours)         |      7B        | 2025 | *73.1*   | *79.0*     | *66.4*     |
> | PerceptionGPT        |     13B        | 2024  | 68.9     | 74.0       | 61.9       |
> | RAS                  |     13B        | 2025  | 75.1     | 80.0       | 70.3       |
> | Text4Seg             |     13B        | 2025  | 73.7     | 78.6       | 67.6       |
> | CoPRS (Ours)         |   13B     | 2025 | **75.5** | **80.3**   | **70.7**   |
>
> ### XDcj-W2
> > Ablation Depth: The GRPO hyperparameter study (Table 4) only tests sampling numbers {2,4,8} without justifying this range. More analysis is needed on how group size affects exploration-exploitation tradeoffs in segmentation contexts.
>
> **Table 4 in §4.4** (original submission) focuses on group sizes $G \in \lbrace 2,4,8 \rbrace$ because larger group size than 8 quickly become impractical for large MLLMs, while $G=1$ essentially collapses GRPO to a non-grouped setting and is less informative. To further deepen this analysis, we additionally ran experiments with $G=16$ on RefCOCO+ and measured the total number of samples required to reach convergence. The detailed results and discussion of trade-offs of group size are provided in our **response to Q2** below.
>
> **Intuitively**, increasing $G$ improves exploration by offering the model to more candidates per image–instruction pair, and sharpening the contrast between good and bad samples. But beyond a certain point (around $G=8$ in our setup) the improvement becomes small while the computational cost grows, which motivates our choice of $G=8$ as a practical trade-off.
>
> ### XDcj-W3
> > Methodological Ambiguity: The heatmap generation process (Eq. 3-4) lacks critical implementation details, e.g., how the MLP maps vision backbone outputs to keys, or why two convolutional layers were chosen for F_fuse. This hinders reproducibility.
>
> Thank you for your attention to the technical details! We added the pseudo-code in **§B.2** in the revised paper explicitly illustrating how the heatmap is generated.
>
> Specifically, For $F_{\text{fuse}}$, we use two stacked 2D convolutional layers to aggregate multi-head transformer scores because this provides a simple, local way to fuse information across heads while preserving spatial structure. Deeper stacks did not bring clear benefits in our preliminary experiments. In addition, we provide **our core model definition** (modeling.py) at the [anonymous link](https://anonymous.4open.science/r/CoPRS-anonymous-meterials/modeling.py), and we will release the full codebase after the review process.

---

> ### Author Response · Authors · 2025-11-25
> **Response to Reviewer XDcj (2/3)**
>
> ### XDcj-W4
>
> > Theoretical Limitations: While empirical correlations are shown, there is no formal analysis of why GRPO’s group-relative advantages are particularly suited for segmentation tasks compared to standard PPO.
>
> We appreciate your query regarding the theoretical basis. While we focus on the empirical efficacy of CoPRS, our choice of GRPO over PPO is motivated by two specific architectural constraints inherent to reasoning segmentation, as discussed in our Appendix:
>
> **Computational Efficiency for End-to-End Training:** Standard PPO requires a separate value function (critic) model of comparable size to the policy, imposing heavy memory and computational costs. Since our framework already integrates a heavy vision backbone and segmentation decoder alongside the MLLM, utilizing GRPO allows us to obviate the need for a critic model. This is crucial for enabling unified, end-to-end optimization of the policy and segmentation modules on standard hardware.
>
> **Suitability for Sparse Rewards:** In our task, the reward (mask quality/IoU) is a sequence-level outcome, not a dense token-level signal. Training a value function that is accurate at each token for such sparse rewards is known to be difficult. GRPO uses the average reward of sampled outputs as the baseline, which naturally fits our "outcome supervision" setup where reasoning and segmentation quality are evaluated holistically.
>
> Our ablation studies in **§4.4** empirically confirm that this combined objective effectively strengthens reasoning while sharpening mask generation.
>
> ### XDcj-Q1
> > How does CoPRS handle ambiguous positional priors in crowded scenes (Figure 5)? The failure cases suggest sensitivity to instance density – could multi-scale features or contrastive learning between instances mitigate this?
>
> This is a very insightful observation about our failure modes in crowded scenes, and it indeed points to a promising direction for future improvements!
>
> As shown in **Fig. 5** (original submission) and discussed in **§C**, CoPRS can produce diffuse positional priors when many similar instances are packed together, and the decoder sometimes struggles to resolve these priors into a single target instance. We see multi-scale features and contrastive learning as natural ways to mitigate this. We plan to further explore such extensions in future work to make CoPRS more robust in dense, same-class scenarios.
>
> ### XDcj-Q2
> > For GRPO sampling numbers (Table 4), what computational overhead does G=8 introduce compared to G=2? A latency/accuracy tradeoff analysis would help practitioners.
>
> Your question about the computational overhead of different group sizes is very relevant. To make this more concrete, **in the table below**, we report the total number of samples needed to reach convergence (loss fluctuation <10% over 300 steps) on RefCOCO+ for $G \in \lbrace 2,4,8,16 \rbrace$, counting all GRPO rollouts per instance.
>
> | Group Size | Val cIoU | TestA cIoU | TestB cIoU | #Samples to Convergence (K) |
> |:-----------|:--------:|:----------:|:----------:|:---------------------------:|
> | 2          | 72.0     | 75.8       | 64.6       | 212.4                        |
> | 4          | 74.7     | 78.5       | 67.9       | 261.5                        |
> | 8 (default)          | 75.9     | 80.3       | 69.7       | 302.5                        |
> | 16         | 76.2     | 80.7       | 69.5       | 394.9                        |
>
> We find that the total sample count **does not grow linearly** with G: larger groups see more candidate trajectories per epoch, which improves exploration and the contrast between good and bad samples. In practice, $G=8$ offers **a favorable  trade-off** between efficiency and performance, where CoPRS achieves clearly stronger performance than $G=2$ while avoiding prohibitive sampling cost. We added the results to **Fig. 6(a)** in the revised paper.
>
> ### XDcj-Q3
> > The correlation analysis (Figure 3) shows association but not causation between heatmaps and masks. Could you disentangle whether performance gains stem primarily from GRPO-enhanced reasoning or the decoder architecture?
>
> This is a very insightful question about where the performance actually comes from.
>
> In our design, the SAM-based decoder architecture is kept fixed across settings and baselines. What CoPRS adds is the MCoT reasoning, the concentration token, and the learned positional prior. In particular, the correlation analysis in Fig. 3 is done **with the decoder parameters frozen** at test time, which indicates that variation in the prior explains a large portion of the mask performance, rather than changes in the decoder. Consistently, **Tab. 3 in §4.4** (original submission)shows that when we remove GRPO, cIoU drops by about 3–5 points on all datasets, suggesting that GRPO-enhanced reasoning and the resulting positional prior make a substantial contribution.

---

> ### Author Response · Authors · 2025-11-25
> **Response to Reviewer XDcj (3/3)**
>
> ### XDcj-S1
> > Add comparisons with RAS-13B and PerceptionGPT-R1 using comparable model sizes.
>
> We appreciate this suggestion.
>
> In our response to **W1**, we present additional experimental results and analysis comparing CoPRS with RAS-13B and PerceptionGPT under comparable backbone settings. We have also added these comparisons into **§4.4** in the revised paper.
>
> ### XDcj-S2
> > Include an ablation on the vision backbone (e.g., ViT-H vs. ViT-L) to clarify performance dependencies.
>
> We followed this suggestion in our additional experiments.
>
> In the table below, we ablate different SAM backbones (ViT-B/L/H) on RefCOCO+ with fixed MLLM backbone Qwen2.5-VL-7B. Larger vision backbones do give slightly better segmentation performance, but the improvement is modest and the overall trend is stable across sizes. Moreover, the vision backbone accounts for only a small fraction of the total parameters compared to the MLLM backbone, so changing its size only slightly increases the overall computational cost.
>
> | Vision Backbone        | #Params(B) | Val cIoU | TestA cIoU | TestB cIoU |
> |:-----------------------|:----------:|:--------:|:----------:|:----------:|
> | SAM-ViT-B              | 8.38       | 73.2     | 77.3       | 67.0       |
> | SAM-ViT-L              | 8.60       | 74.8     | 78.9       | 68.5       |
> | SAM-ViT-H (default)    | 8.93       | **75.9**     | **80.3**       | **69.7**       |
>
> We have included these results and their analysis in **§4.4** in the revised paper.
>
> ### XDcj-S3
> >  Provide pseudocode or extended equations for the heatmap generation process (Section 3.1) to improve reproducibility.
>
> This suggestion is very helpful for improving both clarity and reproducibility. We have added detailed pseudocode for the heatmap generation process at the anonymous link, and we have also included them in **Appendix §B**.
>
> ## XDcj-Rebuttal Potential
> > Addressing the baseline comparison gap (Suggestion 1) and providing theoretical justification for GRPO’s effectiveness in segmentation could significantly strengthen the contribution narrative. Clarifying the heatmap implementation details (Suggestion 3) would enhance methodological rigor.
>
> Thank you for clearly highlighting these important directions for strengthening the paper!
>
> The **baseline comparison** gap is addressed in our response to **W1** and incorporated into **§4.4**. The **GRPO justification** is discussed in our response to **W4**, and the **heatmap implementation** details are clarified in our response to **S3** and incorporated into **Appendix §B**.
>
> We believe these changes follow your guidance and make the contribution and methodology clearer.

---

> ### Author Response · Authors · 2025-11-27
> **A Gentle Reminder of the Post-Rebuttal Feedback**
>
> **Dear Reviewer XDcj,**
>
> Thank you again for your time and valuable feedback on our paper. While we understand that you may have a busy schedule, we would like to confirm whether our responses have addressed your concerns. If you have any further questions or feel that we have misunderstood your feedback, please let us know. We are happy to discuss with you. Furthermore, if our responses and the revised paper have addressed your concerns, we would be grateful if you would consider updating your evaluation.
>
> **Sincerely,**
>
> **The Authors**

---

### Official Review · Reviewer_HRqs · 2025-10-30

**Soundness:** 3
**Presentation:** 3
**Contribution:** 3
**Rating:** 6
**Confidence:** 2

**Summary:**

This paper introduces CoPRS, a novel Multi-modal Chain-of-Thought (MCoT)-based framework that connects language reasoning to visual segmentation through a differentiable positional prior represented as a heatmap. The model integrates a learnable concentration token within a multimodal LLM to aggregate textual reasoning and visual context, producing a heatmap that serves as an interpretable intermediate between reasoning and segmentation. The approach is trained end-to-end by combining Group Relative Policy Optimization (GRPO) for reasoning and supervised segmentation losses for mask prediction.

Empirically, CoPRS achieves state-of-the-art results on the RefCOCO, RefCOCO+, RefCOCOg, and ReasonSeg benchmarks, demonstrating both superior performance and better interpretability compared to prior latent- or text-based reasoning methods. The paper also presents correlation analyses showing that the quality of the learned heatmap aligns strongly with final segmentation accuracy, highlighting the causal link between reasoning and perception.

**Strengths:**

- Overall, writing is clear with intuitive figures.

- The paper introduces an interpretable and differentiable positional prior as an intermediate representation linking language reasoning (via MCoT) to visual segmentation.

- The joint GRPO + supervised objective elegantly integrates reasoning quality and mask precision within a single training loop.

- The heatmap prior provides transparent evidence of where the model is “attending,” supporting qualitative interpretability and quantitative correlation analysis (R > 0.7 between heatmap and mask quality).

- Consistently outperforms both latent reasoning (e.g., LISA, SegLLM) and text-based reasoning methods (e.g., Seg-Zero, Text4Seg) across all RefCOCO variants and ReasonSeg.

- Experimental details are available to reproduce the result.

- The ablation studies on different setups (training mode, reward function, etc) demonstrate further the advantage of the proposed method.

**Weaknesses:**

- The core modules (GRPO, SAM decoder, etc) are existing frameworks; the contribution mainly lies in combining them rather than introducing fundamentally new architectures. Further clarifications on the technical novelty would strengthen the paper.

- Results are tied to Qwen2.5-VL and SAM encoders; unclear how robust the method is across different MLLM backbones or smaller models.

- Although some failure cases are available in the appendix, the paper does not discuss the limitations of the proposed method explicitly in the main text. Adding discussions on limitations will enhance the clarity of the paper.

- GRPO-based multimodal optimization on large MLLMs is computationally heavy; practical feasibility for wider adoption is not discussed.

**Questions:**

- How sensitive is CoPRS to the design of the concentration token prompt (e.g., `<REF_POS>`)?

- Does the GRPO-based reward generalize to new reasoning templates unseen during training? (Does the RL signal capture general reasoning ability rather than just overfit to specific CoT structures?)

- Can the positional prior mechanism transfer to other grounding tasks (e.g., referring tracking or VQA grounding)? (Is the proposed differentiable positional prior is a general paradigm?)

---

> ### Author Response · Authors · 2025-11-25
> **Response to Reviewer HRqs (1/2)**
>
> We sincerely thank you for the constructive feedback. We are encouraged by your positive assessment of our work, particularly your recognition of **the interpretable positional prior**, the elegance of the **joint GRPO objective**, and our **consistent performance gains** over baselines. We also appreciate your feedback regarding the transparency of our **heatmap analysis** and the clarity of the paper's presentation. In the following section, we address your specific concerns point by point.
>
> ### HRqs-W1
> > The core modules (GRPO, SAM decoder, etc) are existing frameworks; the contribution mainly lies in combining them rather than introducing fundamentally new architectures. Further clarifications on the technical novelty would strengthen the paper.
>
> We appreciate the opportunity to clarify our contribution. While we leverage GRPO and the SAM architecture, CoPRS is not a simple assembly. Basically, it introduces a novel **Positional Prior Interface** designed to bridge the gap between high-level reasoning and dense perception, which existing paradigms fail to address effectively.
>
> Our technical novelty consists of two key aspects:
>
> 1. **Differentiable Positional Interface:** Unlike latent methods (opaque features) or text-based methods (discrete coordinates), we design a mechanism where a learnable concentration token explicitly queries image features to generate a dense heatmap. This required designing a specific query head and fusion strategy to translate semantic reasoning into a spatial probability distribution before the final mask decoding.
>
> 2. **Unified Optimization Framework:** We establish a novel training paradigm that couples reinforcement learning (GRPO) on the language path with supervised segmentation on the vision path. This allows the reasoning capability (optimized via reward functions) to be directly aligned with pixel-level concentration in a single end-to-end loop.
>
> The effectiveness of this specific architectural design is evidenced by CoPRS matching or surpassing SOTA on RefCOCO series and ReasonSeg. We have revised our paper to strengthen the presentation of our contributions.
>
> ### HRqs-W2
> > Results are tied to Qwen2.5-VL and SAM encoders; unclear how robust the method is across different MLLM backbones or smaller models.
>
> The concern about the dependence on Qwen2.5-VL and SAM encoders is very reasonable. We examine both the MLLM backbone and the vision backbone in additional experiments.
>
> For the **MLLM backbone**, on RefCOCO+, we report CoPRS versions with both LLaVA-1.5 and Qwen2.5-VL series.  As expected, performance increases with backbone capacity, but the gains across different MLLM backbones are relatively modest. This indicates that CoPRS is **not sensitive to** the specific MLLM architecture and that our improvements largely transfer across different backbone choices.
>
> | Method     | Backbone    | Val cIoU | TestA cIoU | TestB cIoU |
> |:-----------|:------------|:--------:|:----------:|:----------:|
> | CoPRS-3B   | Qwen2.5-VL  | 71.8     | 78.9       | 66.5       |
> | CoPRS-7B   | Qwen2.5-VL  | 75.9     | 80.3       | 69.7       |
> | CoPRS-7B   | LLaVA-1.5   | 73.1     | 79.0       | 66.4       |
> | CoPRS-13B  | LLaVA-1.5   | 75.5     | 80.3       | 70.7       |
>
> For the **vision backbone**, in the table below, we ablate different SAM backbones (ViT-B/L/H) on RefCOCO+ with Qwen2.5-VL-7B. Larger visual backbones yield better performance, but the gains are modest and the trends are stable.  Moreover, changing the vision backbone only slightly affects the total number of parameters, since the MLLM backbone dominates the model size. Therefore varying its size only **slightly increases the computational cost**.
>
> | Vision Backbone        | #Params(B) | Val cIoU | TestA cIoU | TestB cIoU |
> |:-----------------------|:----------:|:--------:|:----------:|:----------:|
> | SAM-ViT-B              | 8.38       | 73.2     | 77.3       | 67.0       |
> | SAM-ViT-L              | 8.60       | 74.8     | 78.9       | 68.5       |
> | SAM-ViT-H (default)    | 8.93       | **75.9**     | **80.3**       | **69.7**       |
>
> We added these results to **§4.4** in the revised paper.
>
> ### HRqs-W3
> > Although some failure cases are available in the appendix, the paper does not discuss the limitations of the proposed method explicitly in the main text. Adding discussions on limitations will enhance the clarity of the paper.
>
> We provide analysis of failure cases in §C. In short, **Fig. 5** (original submission) shows that CoPRS can struggle with tiny objects in very high-resolution images (because the positional prior is computed at $256\times256$ and SAM features at $64\times64$, so very small targets may vanish) and with crowded scenes of many similar instances, where the positional prior can diffuse over multiple candidates.  We added a concise summary of these limitations in **§4.3** of the main text in the revised paper.

---

> ### Author Response · Authors · 2025-11-25
> **Response to Reviewer HRqs (2/2)**
>
> ### HRqs-W4
> > GRPO-based multimodal optimization on large MLLMs is computationally heavy; practical feasibility for wider adoption is not discussed.
>
> Your focus on the computational cost of GRPO is very valuable. To quantify this cost, we report **in the table below** the total number of samples required to reach convergence (loss fluctuation <10% over 300 steps) on RefCOCO+ for group sizes $G \in \lbrace 2,4,8,16 \rbrace$, counting all GRPO rollouts per instance.
>
> | Group Size | Val cIoU | TestA cIoU | TestB cIoU | #Samples to Convergence (K) |
> |:-----------|:--------:|:----------:|:----------:|:---------------------------:|
> | 2          | 72.0     | 75.8       | 64.6       | 212.4                        |
> | 4          | 74.7     | 78.5       | 67.9       | 261.5                        |
> | 8 (default)          | 75.9     | 80.3       | 69.7       | 302.5                        |
> | 16         | 76.2     | 80.7       | 69.5       | 394.9                        |
>
> The total number of samples **does not grow linearly** with $G$, because larger groups expose the model to more diverse candidates per epoch, which improves exploration and the contrast between positive and negative samples. Empirically, this leads to **a good trade-off** at $G = 8$, where CoPRS achieves strong performance without incurring prohibitive sampling cost. We added the results in **§4.4** in the revised paper.
>
> ### HRqs-Q1
> > How sensitive is CoPRS to the design of the concentration token prompt (e.g., `<REF_POS>`)?
>
> The concentration token prompt is not a prompt. In our implementation, `<REF_POS>` is tied with **a fixed special token id** with its own learnable embedding, and the model always conditions on this embedding rather than on the literal text string. The special token is registered, with embedding initialized before training, and the name of the token does not affect training or inference.
>
> ### HRqs-Q2
> > Does the GRPO-based reward generalize to new reasoning templates unseen during training? (Does the RL signal capture general reasoning ability rather than just overfit to specific CoT structures?)
>
> This question reflects a very thoughtful view of the RL component in our framework.
>
> The reward function in CoPRS is **independent of the specific instruction templates** and does not rely on any particular prompt wording. The main term is a mask reward that depends only on segmentation quality, and the auxiliary CoT term only checks coarse format validity. Thus, the RL signal is not scoring specific templates, but whether the reasoning output is usable and leads to accurate masks.
>
> Empirically, as shown in **Tabs. 1 and 2 in §4.2**, we use the same GRPO configuration across benchmarks with quite different instruction styles: short referring expressions in RefCOCO(+/g) and more free-form instructions in ReasonSeg. We observe stable performance, which suggests that the learned policy **is not tied to** specific templates but generalizes across different instruction formats.
>
> ### HRqs-Q3
> > Can the positional prior mechanism transfer to other grounding tasks (e.g., referring tracking or VQA grounding)? (Is the proposed differentiable positional prior a general paradigm?)
>
> We thank the reviewer for this insightful suggestion. We strongly agree that the **Positional Prior Interface** is a generalizable paradigm that extends well beyond segmentation.
>
> As we briefly discussed in the Introduction of our main paper, the unified framework and its positional prior "naturally extend to region concentration tasks such as referring tracking and trajectory prediction". This is because our core mechanism generates a differentiable heatmap representing a dense probability distribution of the target based on reasoning. This heatmap is effectively a universal "concentration" signal that can be utilized for:
>
> - Referring Tracking: By associating heatmaps across frames.
> - VQA Grounding: By using the heatmap as the visual evidence supporting the answer.
>
> While our current work focuses on establishing this paradigm within the challenging scope of **reasoning segmentation**, we view CoPRS as providing a "clean starting point for future work on perception aligned with instructions". We are excited to expand this versatile interface to tracking and VQA grounding in our immediate future research.

---

> ### Author Response · Authors · 2025-11-27
> **A Gentle Reminder of the Post-Rebuttal Feedback**
>
> **Dear Reviewer HRqs,**
>
> Thank you again for your time and valuable feedback on our paper. While we understand that you may have a busy schedule, we would like to confirm whether our responses have addressed your concerns. If you have any further questions or feel that we have misunderstood your feedback, please let us know. We are happy to discuss with you. Furthermore, if our responses and the revised paper have addressed your concerns, we would be grateful if you would consider updating your evaluation.
>
> **Sincerely,**
>
> **The Authors**

---

### Official Review · Reviewer_eH3B · 2025-11-01

**Soundness:** 2
**Presentation:** 3
**Contribution:** 2
**Rating:** 4
**Confidence:** 3

**Summary:**

Free-form Reasoning Segmentation requires coupling language reasoning with visual segmentation. Existing work either uses latent reasoning which offers limited interpretability or text-based reasoning which has limited flexibility. This paper introduces CoPRS, which produces concentration tokens through multimodal chain-of-thought, which are then used to generate heatmaps over the images through attention, and finally producing the segment masks. All components of CoPRS are trained end-to-end using two objectives: a GRPO objective over the output tokens and a supervised segmentation objective over the heatmap and the final predictions. Experiments show that CoPRS achieves competitive performance on RefCOCO and ReasonSeg over existing methods.

**Strengths:**

End-to-end training that allows the model to learn to perform CoT reasoning, generate the query embeddings, and generate the mask simultaneously.

The approach shows strong results across two benchmarks, comparing against an extensive set of existing methods.

The paper is well-written and easy to follow.

**Weaknesses:**

The paper aims to balance interpretability and representational fidelity by generating heatmaps over the images as positional priors. However, this heatmap generation process itself is not interpretable. From my understanding, the training objective encourages the heatmap to match the ground truth masking, which could be seen as a one of the latent reasoning methods discussed in the introduction, and suffers the same downsides of interpretability.

CoPRS relies on a vision backbone to produce the key embeddings that are used to generate the positional prior. This introduces additional computation compared to other methods. Furthermore, the quality of the vision backbone model seems very important, but the paper does not examine the effect of it on the overall performance.

Table 5 and 6 shows ablation on the model’s training objectives. However, there is no in-depth analysis of these results. How are the weights decided?

**Questions:**

I don’t quite see how H_prior can always be visualized as in figure 2. Does H_prior always have the fixed dimension size equal to the image? What does each element in K represent?

How does this differ from latent reasoning method where the latent reasoning is done via an attention mechanism where the attention scores can be extracted (ex. the coder in PixelLM)? Wouldn’t the attention score also be interpreted as heatmaps?

I don’t think it is mentioned in the paper, what is CoPRS an acronym for?

Minor:

Typo: line 297, 406

The citation for SegLLM is incorrect.

---

> ### Author Response · Authors · 2025-11-25
> **Response to Reviewer eH3B (1/2)**
>
> We sincerely thank the reviewer for their time and constructive feedback. We are encouraged that you recognized the novelty of our **end-to-end training framework** for simultaneous CoT reasoning and mask generation, as well as our **strong empirical performance** compared to an extensive set of existing methods. In the following section, we address your specific questions and concerns point-by-point.
>
> ### eH3B-W1
> > The paper aims to balance interpretability and representational fidelity by generating heatmaps over the images as positional priors. However, this heatmap generation process itself is not interpretable. From my understanding, the training objective encourages the heatmap to match the ground truth masking, which could be seen as one of the latent reasoning methods discussed in the introduction, and suffers the same downsides of interpretability.
>
> Thank you for raising this important point regarding the interpretability of the heatmap generation. We clarify our design and provide evidence below:
>
> 1. **Conceptual Comparison: Heatmap as "Explicit Visual Rollout" vs. Latent Reasoning.** We respectfully distinguish our approach from the "latent reasoning" methods discussed in the introduction.
>
>    **(a)** Latent methods typically rely on opaque embedding tokens (like [SEG]) to bridge reasoning and segmentation, which remain "black boxes" to human observers.
>
>    **(b)**  In contrast, our heatmap acts as an **explicit "visual rollout"**, which is a human-readable 2D probability map. It projects the semantic understanding into the spatial domain before the final segmentation, making the intermediate localization step observable and verifiable rather than latent.
>
> 2. **Evidence of Interpretability and Alignment.** To address the concern that the heatmap might simply be mimicking ground truth without linguistic alignment, we provide both quantitative and qualitative evidence:
>
>    **(a)**  **Quantitative Alignment (Gemini-2.5-Flash Eval):** We performed a new experiment using Gemini-2.5-Flash as an independent judge to score the consistency of the generated CoT reasoning. As detailed in the table below and in [anonymous link](https://anonymous.4open.science/r/CoPRS-anonymous-meterials/fig_cot_corr.pdf), we observed a positive correlation between the CoT consistency scores and the *heatmap IoUs*. This suggests that the heatmap is not just matching the ground truth in isolation; it is semantically grounded in the linguistic reasoning quality.
>
>    **(b)**  **Qualitative Inspection:** As shown in **Figs. 4 and 6**  of the paper (original submission), we visualize the full trajectory: *Image → Instruction → CoT → Heatmap → Mask*. These real-world examples allow direct inspection, demonstrating that the heatmap spatially reflects the logic derived in the CoT steps, thereby serving its purpose as an interpretable positional prior.
>
> | CoT Consistency Score | #Samples | Avg. Heatmap IoU | Avg. Mask IoU |
> |:----------------------|:--------:|:----------------:|:-------------:|
> | [0, 0.25)             |   225    |       0.25       |     0.55      |
> | [0.25, 0.5)           |   568    |       0.51       |     0.78      |
> | [0.5, 0.75)           |   749    |       0.69       |     0.90      |
> | [0.75, 1.0]           |   359    |       0.82       |     0.94      |
>
> We added both the figure and table to **§4.3** in the revised paper.
>
> ### eH3B-W2
> > CoPRS relies on a vision backbone to produce the key embeddings that are used to generate the positional prior. This introduces additional computation compared to other methods. Furthermore, the quality of the vision backbone model seems very important, but the paper does not examine the effect of it on the overall performance.
>
> We appreciate your question. As illustrated in Fig. 2 in our main paper, the key embeddings for computing positional prior are **the copy** of encoded image features in the segmentation module. It introduces no extra computation beyond existing methods.
>
> Besides, we did not particularly ablate the vision backbone $\mathcal{F}_\text{enc}$ because it is bound to the segmentation model, which is kept the same among the experiments. We agree that examining the effect of vision backbone size on overall performance is valuable. We report the results in the following table. (Note: *Changing the vision backbone will influence the extraction of the key embeddings as well as the image features in the segmentation module.*)
>
> | Vision Backbone        | #Params(B) | Val cIoU | TestA cIoU | TestB cIoU |
> |:-----------------------|:----------:|:--------:|:----------:|:----------:|
> | SAM-ViT-B              | 8.38       | 73.2     | 77.3       | 67.0       |
> | SAM-ViT-L              | 8.60       | 74.8     | 78.9       | 68.5       |
> | SAM-ViT-H (default)    | 8.93       | **75.9**     | **80.3**       | **69.7**       |
>
> We added the results to **Tab. 5** in the revised paper.

---

> ### Author Response · Authors · 2025-11-25
> **Response to Reviewer eH3B (2/2)**
>
> ### eH3B-W3
> > Table 5 and 6 shows ablation on the model’s training objectives. However, there is no in-depth analysis of these results. How are the weights decided?
>
> We chose these weights based on task experience: (1) **segmentation quality should dominate**, while **the format term acts as a regularizer** to keep outputs parsable, and (2) Dice loss should capture region overlap while Focal loss compensates for foreground–background imbalance.
>
> In Tabs. 5 and 6 (original submission), we adjusted these weights to probe representative settings. The trends confirmed our expectations, though we did not perform the full grid search for optimal values of these weights (given the high expense of LLM experiments).
>
> We added an explicit explanation of these design choices and ablation ranges to **§4.4** in the revised paper.
>
> ### eH3B-Q1
> > I don’t quite see how H_prior can always be visualized as in figure 2. Does H_prior always have the fixed dimension size equal to the image? What does each element in K represent?
>
> Let us clarify them:
>
> 1. $\mathbf{H}_\text{prior}\in\mathbb{R}^{1 \times 256 \times 256}$ is a fixed-size single-channel heatmap. We visualized it similar to Fig. 2 by bilinearly upsampling to match the original image resolution.
>
> 2. $\mathbf{K} \in \mathbb{R}^{C \times H' \times W'}$ (in our case, $256 \times 64 \times 64$) represents the image feature keys derived from the vision backbone (SAM ViT-H), used in the standard attention mechanism. Each element represents the $C$-dimensional spatial feature vector. These keys interact with the queries ($\mathbf{Q}$) via the dot-product operation ($\mathbf{QK}^\top$) to generate the attention weights that form the positional prior.
>
> We clarified this by adding **Algorithm 1** to **§B.2** in the revised paper.
>
> ### eH3B-Q2
> > I don’t think it is mentioned in the paper, what is CoPRS an acronym for?
>
> Sorry for the confusion. CoPRS (pronounced as *co-press*) stands for **Co**T-based **P**ositional perception model for **R**easoning **S**egmentation. We added this explanation in the revised paper.
>
> ### eH3B-Q3
> > Minor.
>
> Thank you for your careful reading and for pointing them out! We have fixed them in the revised version.

---

> ### Author Response · Authors · 2025-11-27
> **A Gentle Reminder of the Post-Rebuttal Feedback**
>
> **Dear Reviewer eH3B,**
>
> Thank you again for your time and valuable feedback on our paper. While we understand that you may have a busy schedule, we would like to confirm whether our responses have addressed your concerns. If you have any further questions or feel that we have misunderstood your feedback, please let us know. We are happy to discuss with you. Furthermore, if our responses and the revised paper have addressed your concerns, we would be grateful if you would consider updating your evaluation.
>
> **Sincerely,**
>
> **The Authors**

---

### Official Review · Reviewer_vsKC · 2025-11-01

**Soundness:** 3
**Presentation:** 3
**Contribution:** 3
**Rating:** 4
**Confidence:** 4

**Summary:**

The paper proposes CoPRS, a CoMT approach that extracts a query from the MLLM. The query is used to generate a differentiable heatmap with cross-attention and acts as a positional prior to the segmentation decoder. The key contributions are:
- A dense, differentiable heatmap is an interpretable link of the MLLM and mask generation.
- A training framework that combines GRPO and supervised objective.
- CoPRS achieves good results on RefCOCO series and ReasonSeg.

**Strengths:**

- The paper addresses a relevant multimodal problem, bridging language reasoning with spatial localization.
- The positional prior offers intermediate interpretability, given the correlation between the quality of the prior and that of the final segmentation mask
- Clear empirical gains on evaluation benchmarks.
- Cleverly combining reinforcement learning with supervised learning

**Weaknesses:**

- The new architecture adds complexity that must be quantified to justify a reduction in speed by the performance gains.
- Although CoT reasoning is emphasized, the actual CoT outputs and their role are not examined. There is no evidence that linguistic reasoning aligns with the generated heatmap.
- Interpretability was studied only through correlation between the heatmap and the final mask, both in the visual domain. The interpretability role of reasoning is unclear.
- GRPO, although more efficient than PPO, is still expensive on a large backbone. The ablation only compares with/without GRPO, without exploring how reasoning reward affects segmentation quality.

**Questions:**

- How sensitive are the reported gains to the MLLM backbone? Ablations with a smaller model or from a  different family (e.g. LLaVa) will show how dependent the model is on the specific architecture.
- In the GRPO reward design, the choice “0.7 for mask and 0.3 for CoT” might be reasonable, but it is left unexplained.
- How correct are the CoT reasoning trajectories, and how do they contribute to interpretability?

---

> ### Author Response · Authors · 2025-11-25
> **Response to Reviewer vsKC (1/2)**
>
> We sincerely thank the reviewer for the time and constructive feedback. We are encouraged that you recognize the relevance of our work in **bridging language reasoning with spatial localization**, as well as the **interpretability** of our positional prior and the **effective combination of RL and SFT**. In the following section, we address your specific concerns point-by-point.
>
> ### vsKC-W1
> > The new architecture adds complexity that must be quantified to justify a reduction in speed by the performance gains.
>
> We appreciate your attention to efficiency. Compared to representative baselines **Seg-Zero** and **Seg-R1** using GRPO, CoPRS only adds a lightweight query head and a small extra computation for positional prior.
>
> The tables below include the performance and efficiency comparisons using the same backbones.
>
> | Method            | #Params(B) | GFLOPs (Inference) | Val cIoU | Test cIoU |
> |:------------------|:----------:|:------------------:|:--------:|:---------:|
> | Seg-R1-3B         |      3.97  |             9096.69 |    56.2 |      46.6 |
> | Seg-Zero-3B       |      3.97  |                 --  |    53.1 |      48.6 |
> | CoPRS-3B (Ours)   |      4.39  |             9551.52 |    **60.6** |      **52.7** |
>
> | Method            | #Params(B) | GFLOPs (Inference) | Val cIoU | Test cIoU |
> |:------------------|:----------:|:------------------:|:--------:|:---------:|
> | Seg-R1-7B         | 8.51       | 20198.96           | 41.2     | 53.7      |
> | Seg-Zero-7B       | 8.51       | 21816.71           | 62.0     | 52.0      |
> | CoPRS-7B (Ours)   | 8.93       | 22283.68           | **64.5**     | **55.1**      |
>
> We can learn that CoPRS achieves substantially better performance under both backbones, with comparable parameter counts and inference cost. We added the results to **§B.4** in the revised paper.
>
> ### vsKC-W2
> > Although CoT reasoning is emphasized, the actual CoT outputs and their role are not examined. There is no evidence that linguistic reasoning aligns with the generated heatmap.
>
> Insightful question. CoPRS inherently aligns linguistic reasoning with the visual output through its training mechanism. Specifically, the CoT trajectory is optimized by outcome rewards computed based on the final decoded masks. This encourages the *linguistic reasoning* to guide the generation of the *final mask*. Besides, **Fig. 3** "Correlation analysis" in the main paper shows the positive correlation between the *generated heatmap* and *decoded masks*. Combining the two cues, we implicitly establish the alignment between the linguistic reasoning and the generated heatmap.
>
> To address this concern more directly, we used Gemini-2.5-Flash API as an independent evaluator. We scored the consistency (range: 0-1, weighted average over four dimensions, following [1]) between the image-instruction pair and the generated CoT on the RefCOCO+ testA split. The resulting scatter plots (see [anonymous link](https://anonymous.4open.science/r/CoPRS-anonymous-meterials/fig_cot_corr.pdf)) and linear correlation analysis confirm a **positive correlation** between the *CoT consistency scores* and the *Heatmap IoUs*. Besides, a group analysis further quantifies this relationship, showing that higher quality CoT outputs correspond to higher average Heatmap IoU. This quantitative analysis directly demonstrates the desired alignment between the CoT and the generated heatmap.
>
> | CoT Consistency Score | #Samples | Avg. Heatmap IoU | Avg. Mask IoU |
> |:----------------------|:--------:|:----------------:|:-------------:|
> | [0, 0.25)             |   225    |       0.25       |     0.55      |
> | [0.25, 0.5)           |   568    |       0.51       |     0.78      |
> | [0.5, 0.75)           |   749    |       0.69       |     0.90      |
> | [0.75, 1.0]           |   359    |       0.82       |     0.94      |
>
> We added the correlation analysis between CoT and segmentation to **§4.3** in the revised paper.
>
> ### vsKC-W3
> > Interpretability was studied only through correlation between the heatmap and the final mask, both in the visual domain. The interpretability role of reasoning is unclear.
>
> Thank you for the question. We address this interpretability concern in two aspects.
>
> First, in our main paper, **Figs. 4 (in §4.3) and 6 (in §C)** (original submission, intuitive and real samples from ReasonSeg) show the image, instruction, full CoT trajectory, and predicted segmentation masks, providing direct inspection of how intermediate reasoning process relates to the positional prior and final prediction.
>
> Second, as just mentioned above in the response to **W2**, we have added correlation analysis between the *CoT (textual domain) consistency* and the *heatmap/mask IoUs*. They supplement extra cues to support that stronger linguistic reasoning facilitates better spatial localization.

---

> ### Author Response · Authors · 2025-11-25
> **Response to Reviewer vsKC (2/2)**
>
> ### vsKC-W4
> > GRPO, although more efficient than PPO, is still expensive on a large backbone. The ablation only compares with/without GRPO, without exploring how reasoning reward affects segmentation quality.
>
> Good advice. We explore this in two aspects.
>
> First, we investigate the impact of **group size**. We report the total number of samples required to reach convergence (loss fluctuation <10% over 300 steps) under group sizes $G \in \lbrace 2,4,8,16 \rbrace$, counting all GRPO rollouts per instance (as shown in the table below). The results are on RefCOCO+.
>
> | Group Size | Val cIoU | TestA cIoU | TestB cIoU | #Samples to Convergence (K) |
> |:-----------|:--------:|:----------:|:----------:|:---------------------------:|
> | 2          | 72.0     | 75.8       | 64.6       | 212.4                        |
> | 4          | 74.7     | 78.5       | 67.9       | 261.5                        |
> | 8 (default)          | 75.9     | 80.3       | 69.7       | 302.5                        |
> | 16         | 76.2     | 80.7       | 69.5       | 394.9                        |
>
> Particularly, sample amount for convergence does not grow linearly with G, because larger groups offer more diverse candidates per step, improving exploration and the contrast between positive and negative samples. Empirically, this yields a good trade-off around $G=8$. We added the results to **Fig. 6(a)** in the revised paper.
>
> Second, we explore the impact of **reward mixing ratio**. By default, we choose a 0.7: 0.3 ratio between segmentation score and format score to *prioritize segmentation accuracy* while still encouraging correct textual outputs. In the main paper, **Tab. 5 in §4.4** (original submission) compares different weightings between the segmentation reward and the CoT format score, and the results support this choice, showing that this ratio makes sense. The result simultaneously shows that training without format score (ratio 1.0 : 0.0) hurts the performance.
>
> ### vsKC-Q1
> > How sensitive are the reported gains to the MLLM backbone? Ablations with a smaller model or from a different family (e.g. LLaVa) will show how dependent the model is on the specific architecture.
>
> Got it. Nice suggestion! Let us present the results on RefCOCO+ dataset.
>
> First, for a more consistent comparison with existing baselines, we train CoPRS versions with **LLaVA-1.5 of 7B and 13B** and compare them with baselines using the same backbones, where **bold** is best in 13B and *italic* is best in 7B.
>
> | Method               | LLaVA-1.5 Size | Year  | Val cIoU | TestA cIoU | TestB cIoU |
> |:---------------------|:--------------:|:-----:|:--------:|:----------:|:----------:|
> | SegLLM               |      7B        | 2025  | 70.3     | 73.0       | 62.5       |
> | UniRES               |      7B        | 2025  | 71.6     | 76.0       | 64.4       |
> | PerceptionGPT        |      7B        | 2025  | 68.5     | 73.9       | 61.3       |
> | Text4Seg             |      7B        | 2025  | 72.1     | 77.6       | 66.1       |
> | CoPRS (Ours)         |      7B        | 2025 | *73.1*   | *79.0*     | *66.4*     |
> | PerceptionGPT        |     13B        | 2024  | 68.9     | 74.0       | 61.9       |
> | RAS                  |     13B        | 2025  | 75.1     | 80.0       | 70.3       |
> | Text4Seg             |     13B        | 2025  | 73.7     | 78.6       | 67.6       |
> | CoPRS (Ours)         |   **13B**      | 2025 | **75.5** | **80.3**   | **70.7**   |
>
> Second, for a comprehensive ablation on backbone, we also report CoPRS versions with both LLaVA-1.5 and Qwen2.5-VL series:
>
> | Method     | Backbone    | Val cIoU | TestA cIoU | TestB cIoU |
> |:-----------|:------------|:--------:|:----------:|:----------:|
> | CoPRS-3B   | Qwen2.5-VL  | 71.8     | 78.9       | 66.5       |
> | CoPRS-7B   | Qwen2.5-VL  | **75.9**     | **80.3**      | 69.7       |
> | CoPRS-7B   | LLaVA-1.5   | 73.1     | 79.0       | 66.4       |
> | CoPRS-13B  | LLaVA-1.5   | 75.5     | **80.3**       | **70.7**       |
>
> As expected, performance scales with backbone capacity, but in both 7B and 13B settings CoPRS consistently outperforms methods using the same LLaVA-1.5 backbones, confirming that the gains are not tied to a specific architecture and that our improvements are complementary to backbone strength. We added the results to **Tab. 4** in the revised paper.
>
> ### vsKC-Q2
> > In the GRPO reward design, the choice “0.7 for mask and 0.3 for CoT” might be reasonable, but it is left unexplained.
>
> We choose this default ratio based on empirical experience, so that the reward emphasizes segmentation quality while still accounting for CoT format. The sensitivity analysis in Tab. 5 in §4.4 (original submission)
>  supports that this choice is appropriate.
>
> ### vsKC-Q3
> > How correct are the CoT reasoning trajectories, and how do they contribute to interpretability?
>
> Please refer to the answers to **W2** and **W3**. :)

---

> ### Author Response · Authors · 2025-11-27
> **A Gentle Reminder of the Post-Rebuttal Feedback**
>
> **Dear Reviewer vsKC,**
>
> Thank you again for your time and valuable feedback on our paper. While we understand that you may have a busy schedule, we would like to confirm whether our responses have addressed your concerns. If you have any further questions or feel that we have misunderstood your feedback, please let us know. We are happy to discuss with you. Furthermore, if our responses and the revised paper have addressed your concerns, we would be grateful if you would consider updating your evaluation.
>
> **Sincerely,**
>
> **The Authors**

---

### Author Response · Authors · 2025-11-25
**General Responses to all Reviewers**

**Dear Chairs and Reviewers,**
Thank you for the constructive and detailed feedback on our submission.
This note collects general responses to concerns that appeared across multiple reviews. We organize them into four short themes before the point-by-point responses.

**1. Clarifying Interpretability (vsKC, eH3B)**
Reviewers asked whether our positional prior is truly more interpretable than latent feature connectors, since it is also trained to match ground-truth masks.
Conceptually, latent reasoning methods pass opaque embedding tokens (e.g., [SEG]) from the LLM to the decoder, whereas CoPRS exposes a 2D heatmap over image coordinates as an intermediate object: it can be directly visualized, inspected against the instruction, and compared to the final mask.
Empirically, §4.3 now provides quantitative evidence that (i) the heatmap and decoded mask are strongly correlated during both training and inference (Fig. 3 in the revised paper), and (ii) an external CoT consistency score correlates positively with both heatmap IoU and mask IoU (Tab. 3 and Fig. 4 in the revised paper). Together with the visual trajectories (image → instruction → CoT → heatmap → mask) shown in our qualitative figures, these results support that the heatmap acts as an interpretable positional prior that is aligned with the linguistic reasoning, rather than a purely latent surrogate.

**2. Performance Across MLLM Backbones (vsKC, HRqs)**
To disentangle backbone effects from our contribution, we conduct an explicit MLLM ablation on RefCOCO+ (Tab. 4 in the revised paper). We run CoPRS with both LLaVA-1.5-7B/13B and Qwen2.5-VL-3B/7B under the same training setup. As shown in Tab. 4, performance scales with backbone capacity, but the gains across different MLLMs are modest and the overall trend is stable across sizes. This supports that our improvements are not tied to a particular MLLM architecture and are largely complementary to backbone strength.

**3. GRPO Performance and Efficiency (vsKC, HRqs, XDcj)**
Regarding the balance between performance and the cost of GRPO, the revised paper now analyzes group size in Fig. 6(a), jointly reporting performance and the number of samples needed for convergence. Increasing the group size improves accuracy while the effective sample cost grows sub-linearly, since each update sees more diverse candidates. In practice, we find G=8 offers a good trade-off between performance and efficiency, indicating that GRPO is computationally feasible in our setting rather than prohibitively expensive.

**4. Effect of Vision Backbone Choice (eH3B, HRqs, XDcj)**
In Tab. 5 (revised paper), we ablate SAM backbones (ViT-B/L/H) on RefCOCO+ with a fixed Qwen2.5-VL-7B. Larger vision backbones bring slightly better segmentation performance, but the gains are modest and the trends remain stable. Since the MLLM dominates the total parameters, changing the vision backbone only marginally affects model size and computational cost, indicating that our conclusions are not sensitive to this choice.

We hope these clarifications help address the main common concerns and better contextualize our design and empirical results.
We sincerely appreciate your time and thoughtful comments.

**Sincerely,**
**The Authors**

---

### Author Response · Authors · 2025-11-25
**Summary of Revisions to the Paper**

**Dear Chairs and Reviewers,**

We appreciate your constructive feedback and the opportunity to revise our paper.
This document summarizes the main changes made in the revised version.
Separate responses will provide global clarifications and point-by-point responses to individual comments.

**Below we list the key revisions:**

1. We update the abstract and introduction to more clearly describe the correlation studies.
2. We clarify what “CoPRS” stands for in §1.
3. We refine and expand the research questions in §4.
4. We highlight our method in Tabs. 1 and 2 by a gray background in §4.2.
5. We add a new paragraph titled *Correlation between CoT and Segmentation Quality* in §4.3.
6. We merge the discussion on *Correlation between Heatmap and Mask* in §4.3.
7. We add a concise summary of failure cases in §4.3.
8. We introduce ablations over varied MLLM backbones in §4.4.
9. We ablate different vision backbones in §4.4.
10. We decouple the GRPO hyperparameter ablation in §4.4 into two parts: group size and reward coefficient.
11. We convert the ablation tables into four subfigures for better visualization, presented together in Fig. 6 in §4.4.
12. We provide a more in-depth analysis of Fig. 6(c)(d), explaining how the weights are chosen in §4.4.
13. We update the Reproducibility Statement to add design details.
14. We explicitly specify the reward function design in Appendix §B.2.
15. We add pseudo-code to clarify the heatmap generation in Appendix §B.2.
16. We provide a quantitative analysis of inference efficiency in Appendix §B.4.
17. We correct typos throughout the paper.

We hope these revisions help clarify our method and address the main concerns raised in the reviews.
Thank you again for your time and thoughtful comments.

**Sincerely,**

**The Authors**

---

### Author Response · Authors · 2025-11-28
**Welcome to comment during the OpenReview API security incident!**

**Dear Reviewers and Chairs,**

We were sorry to learn that OpenReview has just suspended review editing privileges due to the recent API security incident. We understand that, as a result, you may currently be unable to update your official reviews.

Fortunately, we find the comment function remains active. We would be very happy to clarify any remaining questions or discuss any points that remain unclear. We would greatly appreciate the reviewers' input and evaluation of our added results and improvements to the paper.

We also wish you the best of luck with your own submissions and hope everything proceeds smoothly for you.

**Sincerely,**

**The Authors**

---

### Author Response · Authors · 2025-12-02
**Summary of Rebuttal Phase**

**Dear Chairs,**

We sincerely appreciate the time and effort you have dedicated to overseeing the review process. We have explicitly addressed all reviewer comments and uploaded a revised manuscript reflecting these updates. To assist you in efficiently assessing the rebuttal and to facilitate an informed decision, below is a concise summary of how our revisions reinforce the **strengths** of CoPRS and resolve the **key concerns** raised by reviewers.

1. Core Strengths & Contributions
   - **Validated Interpretability:** Unlike methods that pass latent embeddings to decoders, CoPRS exposes a **differentiable heatmap** as an interpretable intermediate. In **§4.3**, we provide quantitative evidence of **strong positive correlations** among the CoT trajectory, the Heatmap, and the Mask.  This proves our heatmap is not a latent surrogate but a true semantic bridge, enabling transparent trajectories: **Image $\to$ CoT $\to$ Heatmap $\to$ Mask**.
   - **Unified Training:** We successfully integrate GRPO (RL) with segmentation supervision, proving that the unified objective significantly enhances reasoning segmentation.
   - **Solid Experiments:** We conducted a comprehensive evaluation to validate our method, covering **SOTA results** on benchmarks, rigorous **correlation** analysis, qualitative **visualizations**, detailed **ablation** studies, and **efficiency** comparison.

2. Resolution of Key Concerns

    | Reviewer | Highlighted strengths | Key concerns | Our response |
    |:---------|:-------------------------|:----------------------|:----------------------------------------------------|
    | **vsKC (4)** |  • **Soundness:** Interpretable positional prior |  1. CoT correlation  |  1. Added **Correlation Analysis** between CoT and heatmap/mask (Tab. 3 & Fig. 4). |
    | | • **Effectiveness:** Clear empirical gains | 2. Efficiency | 2. Added **Params/GFLOPs** Tables (Tabs. 6 & 7) and **group size** ablation (Fig. 6a). |
    | | • **Novelty:** RL-supervised learning combination | 3. MLLM backbone | 3. Validated on **LLaVA-1.5/Qwen2.5-VL** (Tab. 4). |
    | | | 4. Reward design | 4. Added clarification about reward design (§4.4). |
    | **eH3B (4)** | • **Novelty:** End-to-end training framework | 1. Interpretability | 1. Extended **quantitative analysis** (§4.3); clarified **qualitative inspection** (Figs. 4 and 6) |
    | | • **Effectiveness:** Strong benchmark results | 2. Vision backbone | 2. Ablated vision backbones: **SAM-B/L/H** (Tab. 5). |
    | | | 3. Ablated weights | 3. Added an explicit explanation of the design choices (§4.4). |
    | **HRqs (6)** | • **Novelty:** Interpretable positional prior | 1. Backbone effect | 1. Ablated **MLLM/Vision backbones** (Tab. 4 & 5). |
    | | • **Soundness:** Transparent interpretability evidence | 2. GRPO efficiency | 2. Extended **efficiency ablation** on GRPO group size (Fig. 6a). |
    | | • **Effectiveness:** Superior segmentation performance | 3. Technical novelty | 3. Clarified the novelty of Differentiable Positional **Interface** and **Unified Framework** (§1). |
    | **XDcj (6)** | • **Novelty:** Creative MCoT integration | 1. Heatmap detail | 1. Added **pseudo-code** of heatmap generation (Alg. 1). |
    | | • **Significance:** Addresses interpretability gap | 2. Backbone effect | 2. Added **ablations** on MLLM backbone and vision backbone (Tab. 4 & 5). |
    | | • **Effectiveness:** Significant performance improvement | 3. GRPO efficiency | 3. Extended ablation on group size (Fig. 6a) and clarified the **trade-off** (§4.4). |

The revised paper now provides concrete empirical evidence supporting the interpretability and efficiency of CoPRS. We believe these updates fully address the reviewers' reservations and confirm the paper's readiness for publication.

**Sincerely,**

**The Authors**

---

### Meta-Review · Area_Chair_2cZY · 2026-01-07

**Summary:**

The author rebuttal has addressed most of the raised concerns The newly added experiments significantly strengthen the paper and enhance its overall value. As a result, the submission appears to be a clear accept. The only remaining concern comes from Reviewer eH3B, which I believe is philosophically well founded. This concern points to a degree of over-claiming in the paper that the authors should address.

**Reviewer Concerns:**

As listed in the summary.

**Reviewer Scores:**

I think all the reviewers would change their scores. Despite that the three of them have already given positive scores while point out some limitations like novelty issue, they may decide to keep the scores.

---

### Decision · Program_Chairs · 2026-01-26

Accept (Poster)